# Characterization of the mechanism by which the RB/E2F pathway controls expression of the cancer genomic DNA deaminase APOBEC3B

Pieter A Roelofs[1,2], Chai Yeen Goh[3†], Boon Haow Chua[3,4†], Matthew C Jarvis[1], Teneale A Stewart[1,5], Jennifer L McCann[1,6], Rebecca M McDougle[1,7], Michael A Carpenter[1,6], John WM Martens[8], Paul N Span[2], Dennis Kappei[3,4], Reuben S Harris[1,6]*

[1]Department of Biochemistry, Molecular Biology and Biophysics, Masonic Cancer Center, Institute for Molecular Virology, Center for Genome Engineering, University of Minnesota, Minneapolis, United States; [2]Department of Radiation Oncology, Radboud University Medical Center, Nijmegen, Netherlands; [3]Cancer Science Institute of Singapore, National University of Singapore, Singapore, Singapore; [4]Department of Biochemistry, Yong Loo Lin School of Medicine, National University of Singapore, Singapore, Singapore; [5]Mater Research Institute, The University of Queensland, Faculty of Medicine, Brisbane, Australia; [6]Howard Hughes Medical Institute, University of Minnesota, Minneapolis, United States; [7]Hennepin Healthcare, Minneapolis, United States; [8]Erasmus MC Cancer Institute, Erasmus University Medical Center, Rotterdam, Netherlands

*For correspondence:
rsh@umn.edu

†These authors contributed equally to this work

**Abstract** APOBEC3B (A3B)-catalyzed DNA cytosine deamination contributes to the overall mutational landscape in breast cancer. Molecular mechanisms responsible for *A3B* upregulation in cancer are poorly understood. Here we show that a single E2F cis-element mediates repression in normal cells and that expression is activated by its mutational disruption in a reporter construct or the endogenous *A3B* gene. The same E2F site is required for *A3B* induction by polyomavirus T antigen indicating a shared molecular mechanism. Proteomic and biochemical experiments demonstrate the binding of wildtype but not mutant E2F promoters by repressive PRC1.6/E2F6 and DREAM/E2F4 complexes. Knockdown and overexpression studies confirm the involvement of these repressive complexes in regulating A3B expression. Altogether, these studies demonstrate that *A3B* expression is suppressed in normal cells by repressive E2F complexes and that viral or mutational disruption of this regulatory network triggers overexpression in breast cancer and provides fuel for tumor evolution.

## Introduction

Cancer is a collection of diseases characterized by a complex array of mutations ranging from gross chromosomal abnormalities to single-base substitution (SBS) mutations. Over the last decade, analyses of thousands of tumor genome sequences have confirmed this complexity and also, importantly, revealed common patterns or signatures indicative of the sources of DNA damage that led to these observed mutations (most recent pan-cancer analysis by *Alexandrov et al., 2020*; reviewed by *Helleday et al., 2014*; *Roberts and Gordenin, 2014*; *Swanton et al., 2015*; *Venkatesan et al., 2018*). One of the most prominent SBS mutation signatures to emerge is attributable to members of

the APOBEC family of single-stranded (ss)DNA cytosine deaminases (*Alexandrov et al., 2013*; *Burns et al., 2013a*; *Burns et al., 2013b*; *Nik-Zainal et al., 2012*; *Roberts et al., 2013*). Breast, lung, head/neck, cervical, and bladder cancers often have strong APOBEC signatures and subsets of other cancer types have weaker APOBEC contributions. The APOBEC mutation signature consists of C-to-T transitions and C-to-G transversions occurring at cytosine nucleobases in 5'-TCW motifs (W = A or T; SBS2 and SBS13), respectively (*Alexandrov et al., 2020*; *Alexandrov et al., 2013*; *Nik-Zainal et al., 2016*).

The human APOBEC family has nine active family members: APOBEC1, AID, and APOBEC3A/B/C/D/F/G/H (reviewed by *Green and Weitzman, 2019*; *Harris and Dudley, 2015*; *Ito et al., 2020*; *Olson et al., 2018*; *Silvas and Schiffer, 2019*; *Simon et al., 2015*; *Siriwardena et al., 2016*). Although several APOBEC3s have been implicated in cancer mutagenesis including APOBEC3A (A3A) and APOBEC3H (A3H) (*Chan et al., 2015*; *Nik-Zainal et al., 2014*; *Starrett et al., 2016*; *Taylor et al., 2013*), a particularly strong case can be made for APOBEC3B (A3B). First, A3B is over-expressed in a large fraction of tumors (*Burns et al., 2013a*; *Burns et al., 2013b*; *Ng et al., 2019*; *Roberts et al., 2013*). Second, A3B is the only deaminase family member localizing to the nuclear compartment (*Bogerd et al., 2006*; *Burns et al., 2013a*; *Lackey et al., 2012*; *Lackey et al., 2013*; *Pak et al., 2011*; *Salamango et al., 2018*; *Stenglein et al., 2008*). Third, A3B overexpression triggers strong DNA damage responses and overt cytotoxicity (*Burns et al., 2013a*; *Nikkilä et al., 2017*; *Serebrenik et al., 2019*; *Taylor et al., 2013*; *Yamazaki et al., 2020*). Fourth, *A3B* expression correlates positively with APOBEC signature mutation loads in breast cancer (*Burns et al., 2013a*), and its overexpression associates with branched evolution in breast and lung cancer (*de Bruin et al., 2014*; *Lee et al., 2019*; *Roper et al., 2019*). Fifth, *A3B* expression is induced by human papillomavirus (HPV) and polyomavirus (PyV) infections, which relates to the fact that cervical, head/neck, and bladder cancers have high proportions of APOBEC signature mutations (*Gillison et al., 2019*; *Henderson et al., 2014*; *Starrett et al., 2019*; *Verhalen et al., 2016*; *Vieira et al., 2014*). Last, *A3B* overexpression associates with poor clinical outcomes including drug resistance and metastasis (*Glaser et al., 2018*; *Law et al., 2016*; *Serebrenik et al., 2020*; *Sieuwerts et al., 2017*; *Sieuwerts et al., 2014*; *Walker et al., 2015*; *Xu et al., 2015*; *Yamazaki et al., 2019*; *Yan et al., 2016*). However, in a different subset of cancer types, A3B has been shown to exert genotoxic stress that sensitizes tumor cells to DNA damaging chemotherapies (*Glaser et al., 2018*; *Serebrenik et al., 2020*).

The importance of A3B in cancer mutagenesis has stimulated interest in understanding the mechanisms by which this DNA mutator becomes overexpressed in tumors. A variety of stimuli have been shown to trigger transcriptional upregulation of endogenous *A3B* including small molecules, DNA damaging agents, and viral infections. Phorbol myristic acid (PMA) and lymphotoxin-β induce *A3B* by activating the protein kinase C (PKC) and non-canonical (nc)NF-κB signal transduction pathways (*Leonard et al., 2015*; *Lucifora et al., 2014*). Canonical NF-κB activation also leads to *A3B* upregulation (*Maruyama et al., 2016*) suggesting a mechanistic linkage between inflammatory responses and cancer mutagenesis. Various DNA damaging agents also stimulate *A3B* expression including hydroxyurea, gemcitabine, aphidicolin, and camptothecin (*Kanu et al., 2016*; *Yamazaki et al., 2020*). Interestingly, as alluded above, HPV infection induces *A3B* expression by mechanisms requiring the viral E6 and E7 oncoproteins (*Mori et al., 2015*; *Mori et al., 2017*; *Starrett et al., 2019*; *Verhalen et al., 2016*; *Vieira et al., 2014*; *Warren et al., 2015*; *Westrich et al., 2018*). E6 appears to induce *A3B* in part by recruiting the transcription factor TEAD4 to promoter sequences (*Mori et al., 2015*; *Mori et al., 2017*). JC and BK PyV upregulate *A3B* transcription by a mechanism requiring the LxCxE motif of the viral large T antigen (TAg; *Starrett et al., 2019*; *Verhalen et al., 2016*). HPV E7 also has a LxCxE motif suggesting a shared mechanism in which these viral oncoproteins may activate *A3B* transcription by antagonizing the canonical retinoblastoma tumor suppressor protein RB1 and the related pocket proteins RB-like 1 (RBL1) and RBL2 (reviewed by *An et al., 2012*; *Bellacchio and Paggi, 2013*; *DeCaprio, 2014*; *DeCaprio and Garcea, 2013*; *Rashid et al., 2015*). Viral inactivation of RB1 and RBL1/2 alters interactions with cellular E2F transcription factors and contributes to an accelerated cell cycle with dampened checkpoints. The RB/E2F axis is also frequently disrupted in non-viral cancers such as breast cancer, HPV-negative head/neck cancer, and lung cancer (*Cancer Genome Atlas Network, 2012*; *Ertel et al., 2010*; *Nik-Zainal et al., 2016*; *Cancer Genome Atlas Network, 2015*).

Central to the human RB/E2F axis are eight distinct E2F transcription factors (reviewed by *Cao et al., 2010*; *Fischer and Müller, 2017*; *Sadasivam and DeCaprio, 2013*). E2F1, E2F2, and E2F3 bind target promoters and recruit additional activating proteins to stimulate the expression of cell cycle genes during G1/S. RB1 binds the transactivation domain of these E2Fs and thereby prevents the recruitment of transcription activating factors. E2F4 and E2F5 form complexes with RBL1 or RBL2 and further associate with the MuvB complex, which includes LIN9, LIN37, LIN52, LIN54, and RBBP4. This bipartite assembly, known as the DREAM complex, represses transcription during the G0 and early G1 phases of the cell cycle (*Litovchick et al., 2011*; *Litovchick et al., 2007*; *Pilkinton et al., 2007*). Endogenous Cyclin/CDK complexes, as well as HPV E7 and PyV TAg through LxCxE motifs, dissociate RBL1 and RBL2 from E2F4 and E2F5 and thereby activate transcription (reviewed by *An et al., 2012*; *Bellacchio and Paggi, 2013*; *DeCaprio, 2014*; *DeCaprio and Garcea, 2013*; *Rashid et al., 2015*). E2F6, E2F7, and E2F8 exert their repressive function independent of RB1, RBL1, and RBL2 (*Christensen et al., 2005*; *de Bruin et al., 2003*; *Trimarchi et al., 1998*). E2F6 functions in the Polycomb Repressive Complex (PRC)1.6 complex to repress gene expression during G1-S (*Qin et al., 2012*; *Scelfo et al., 2019*; *Stielow et al., 2018*). The PRC1.6 complex consists of MGA, L3MBTL2, PCGF6, WDR5, E2F6, and TFDP1 (among other proteins), and directly binds DNA through MGA, L3MBTL2, and E2F6 (*Stielow et al., 2018*). Finally, E2F7 and E2F8 repress genes through the S-phase and prevent gene reactivation during the next cell cycle (*Cuitiño et al., 2019*).

Our previous studies showed that *A3B* expression is low in normal tissues (*Burns et al., 2013a*; *Refsland et al., 2010*) and inducible upon PyV TAg expression (*Starrett et al., 2019*; *Verhalen et al., 2016*). *A3B* induction by TAg may occur through the RB/E2F axis, as alluded above, or through a different LxCxE-dependent mechanism. The feasibility of such an alternative mechanism is supported by evidence that LxCxE is a common motif for protein-protein interactions and that HPV E7 uses this motif to bind >100 cellular proteins in addition to RB1, RBL1, and RBL2 (*White et al., 2012*). Here a series of molecular, biochemical, proteomic, and genomic approaches are used to distinguish between these molecular mechanisms. The combined results demonstrate the functionality of a single E2F binding site in the *A3B* promoter and reveal overlapping roles for both E2F4-based DREAM and E2F6-based PRC1.6 complexes in repressing *A3B* transcription in non-tumorigenic cells. Loss of this *A3B* repression mechanism in tumor cells is likely to promote cancer mutagenesis.

## Results

### The *A3B* promoter contains a repressive transcriptional element

To study the mechanism of *A3B* transcriptional regulation, a 950 bp region spanning the *A3B* transcription start site (TSS) was cloned upstream of a firefly luciferase reporter (i.e. −900 to +50 relative to the +1 of the *A3B* TSS; *Figure 1A*). In MCF10A normal-like breast epithelial cells and MCF7 breast cancer cells, which both express low levels of *A3B* (*Burns et al., 2013a*), this construct supported modest levels of transcription activity above those of a promoter-less vector (compare black bars of pGL3-basic versus pA3B-luciferase in *Figure 1B*). Interestingly, similar to upregulation of the endogenous *A3B* gene in our previous studies (*Starrett et al., 2019*), transcription of the *A3B-luciferase* reporter was induced strongly in cells co-expressing the BK PyV truncated T antigen (tTAg) but not in cells co-expressing a LxCxE mutant tTAg (*Figure 1B*).

The JASPAR database (*Fornes et al., 2020*) was then used to predict transcription factor binding sites within the −900 to +50 *A3B* promoter region. This analysis yielded dozens of candidate sites including five putative E2F binding sites (labeled A-E in *Figure 1A*). The functionality of each E2F binding site was assessed by constructing site-directed mutant clusters and comparing *A3B-luciferase* reporter activity in MCF10A and MCF7 (*Figure 1B–C*). Clustered base substitution mutations in sites A, B, and C had negligible effects on basal or tTAg-induced levels of luciferase reporter expression. Clustered mutations in site D caused a two- to three-fold reduction in both basal and tTAg-induced levels of luciferase reporter expression. However, clustered mutations in site E, located at +21 to +28 relative to the TSS, caused a strong five-fold induction of *A3B-luciferase* reporter activity that could not be further increased by tTAg co-expression. Mutations in site E were also epistatic to those in site D, suggesting that site E may be the dominant regulatory site. The importance of site E was confirmed by analyzing additional mutation clusters, which partly or fully spanned site E and

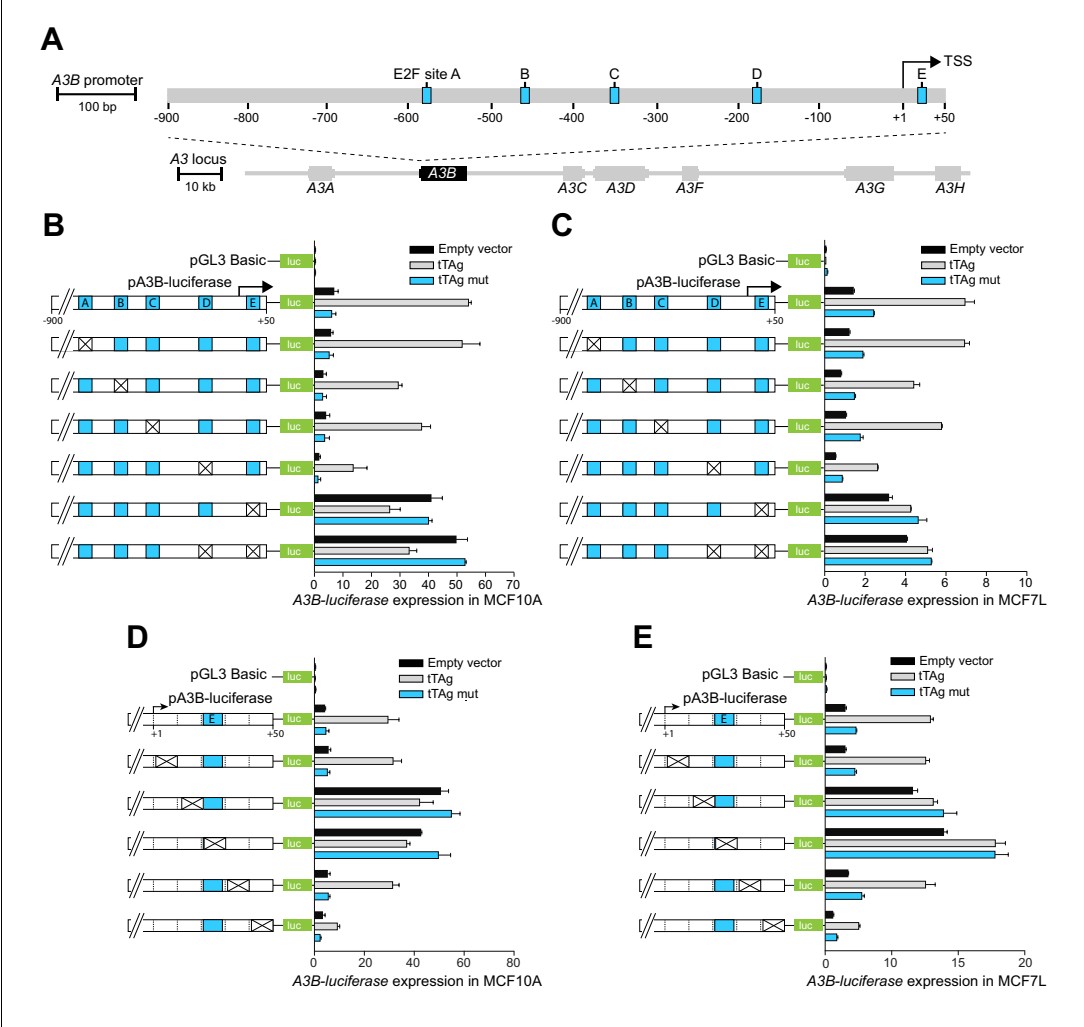

**Figure 1.** The *A3B* promoter harbors a repressive *cis*-element in the +1 to +50 region. (**A**) Schematic of the 7-gene human *APOBEC3* locus with the *A3B* promoter magnified to depict five predicted E2F binding sites (A-E in blue) relative to the TSS at +1 (scales indicated). (**B–E**) Relative luciferase activity of MCF10A or MCF7 cells expressing the indicated firefly luciferase construct (pGL3-basic, pA3B-luciferase, or mutant pA3B-luciferase), a renilla luciferase internal control plasmid (not shown), and a tTAg plasmid (empty, wildtype, or LxCxE mutant). Mutation clusters are depicted by X's (mutant sequences in *Supplementary file 4*). Experiments report mean ± SD of n ≥ 2 technical replicates and are representative of n = 3 biologically independent replicates.

resulted in complete de-repression of *A3B-luciferase* expression (*Figure 1D–E*). Mutation clusters +12 to +20 and +22 to +30 guided additional analyses including proteomics experiments below. Taken together, these results suggested that the +12 to +30 region of the *A3B* promoter including site E is normally bound by a repressive factor and different mutations prevent repression and allow high levels of transcription.

## *A3B* promoter phylogenetic analyses delineate conserved CHR and E2F sites

To gain additional insights into the possible involvement of an E2F complex in *A3B* transcriptional repression, TCGA breast cancer RNA-seq data sets were used to identify 114 genes with expression profiles positively associating with *A3B* (Spearman's rho ≥0.5; n = 1,097 RNA-seq data sets; *Supplementary file 1*). Remarkably, 87% of these genes were shown to be bound by repressive E2F complexes suggesting a common regulatory mechanism (*Litovchick et al., 2007*; *Müller et al., 2014*; *Supplementary file 1*). For instance, *A3B* mRNA levels across primary breast cancer associated strongly with expression levels of *MELK* and *FOXM1* (*Figure 2A*), which both have well-

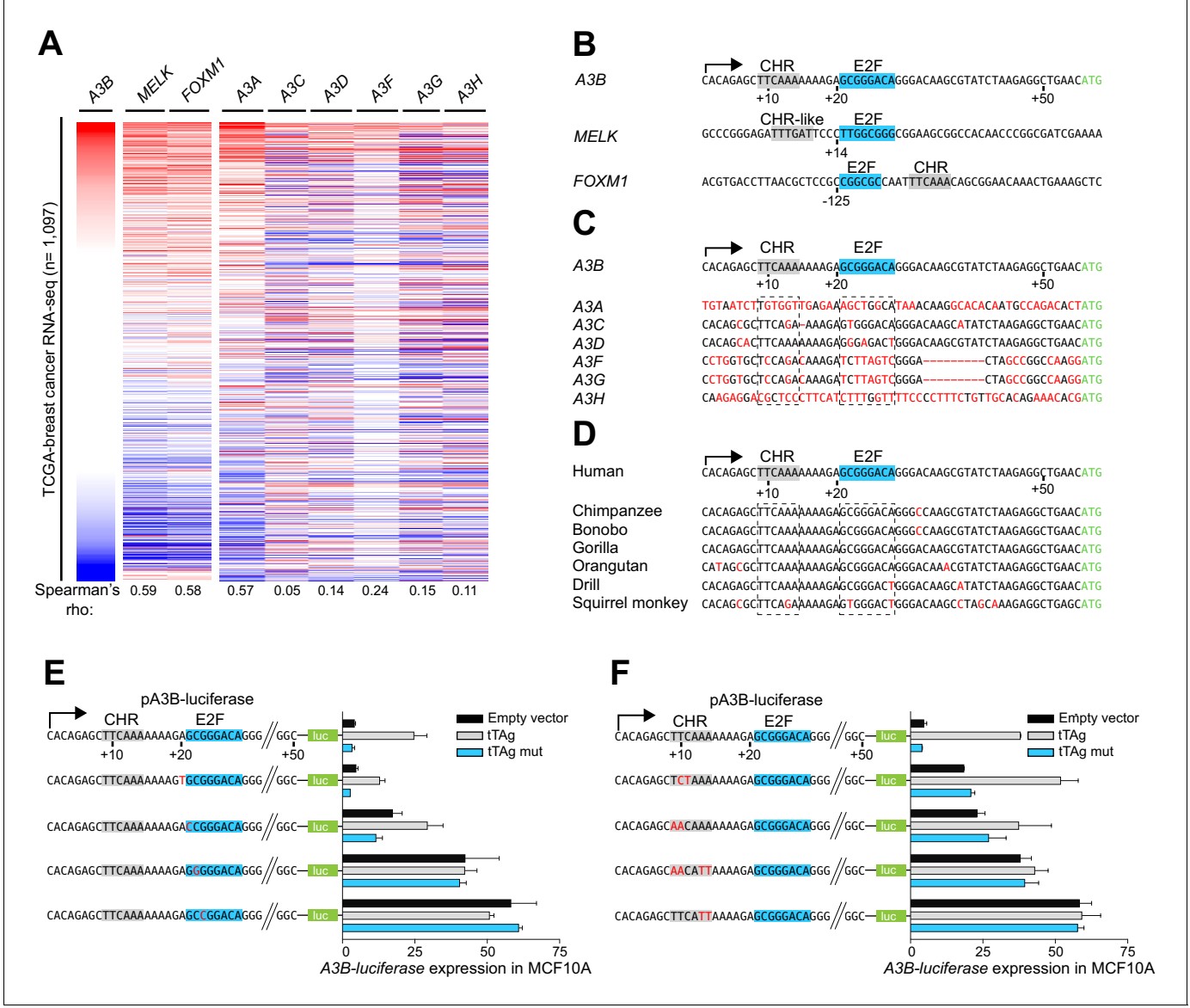

**Figure 2.** *A3B* repression requires both CHR and E2F *cis*-elements. (A) Heatmap depicting high-to-low *A3B* expression levels in TCGA breast cancer specimens (n = 1,097) and correlations with two known RB/E2F target genes, *MELK* and *FOXM1*, and related *APOBEC3* genes (Spearman's rho indicated). (B) Comparison of the *A3B* promoter and analogous regions of *MELK* and *FOXM1*. Known and predicted E2F and CHR elements are indicated in blue and light gray, respectively. (C–D) Alignments of the *A3B* promoter sequence and corresponding promoter sequences of related human *APOBEC3* genes and representative non-human primate *A3B* genes. (E–F) Relative luciferase activity of MCF10A cells expressing the indicated firefly luciferase construct (pA3B-luciferase or mutant pA3B-luciferase), a renilla luciferase internal control plasmid (not shown), and a tTAg plasmid (empty, wildtype, or LxCxE mutant). Panel (E) reports data for E2F mutants and (F) for CHR mutants. Experiments report mean ± SD of n ≥ 2 technical replicates and are representative of n = 3 biologically independent replicates.

described E2F-dependent repression mechanisms (*Litovchick et al., 2007*; *Müller et al., 2017*; *Müller et al., 2014*; *Verlinden et al., 2005*). A subset of these coordinately expressed genes also has a predicted consensus (or near-consensus) cell cycle gene homology region (CHR) element adjacent to the predicted E2F binding site (*Figure 2B* and *Supplementary file 1*). When juxtaposed, these two elements cooperatively facilitate the binding of repressive E2F complexes and suppress gene expression (*Müller et al., 2012*; *Müller et al., 2017*; *Müller et al., 2014*) and reviewed by *Fischer and Müller, 2017*; *Sadasivam and DeCaprio, 2013*. Interestingly, in the *A3B* promoter, both the predicted CHR (+9 to +14) and E2F (+21 to +28) elements occur within the +12 to +30 region defined above in mutagenesis experiments (*Figure 1D–E*).

The global profile of *A3B* mRNA expression in primary breast cancer is distinct from related *A3* genes except for *A3A* (*Figure 2A*). This is explained by differences at potentially critical nucleobase positions in both the CHR and E2F sites in the individual *A3* gene promoters including the most closely related *A3C* promoter region (*Figure 2C* and see below). The *A3A* promoter shares no obvious homology and the associated expression profiles cannot be explained mechanistically at this time. Sequence comparisons with other primates demonstrate that this region of the *A3B* promoter, including juxtaposed CHR and E2F elements, is conserved in hominids and Old World monkeys (*Figure 2D*). Thus, adjacent CHR and E2F sites in the *A3B* promoter are unique amongst *A3* genes, specific to humans and other higher primates, and likely linked to the aforementioned expression patterns.

To interrogate the functionality of the E2F and CHR elements, the *A3B-luciferase* reporter was subjected to additional rounds of site-directed mutagenesis and analysis in MCF10A. Altering the nucleobase immediately 5' of the predicted E2F binding site (+20 A-to-T) had no effect, and changing the first nucleobase of the predicted E2F binding site (+21 G-to-C) caused slight reporter activation but did not affect tTAg inducibility (*Figure 2E*). In contrast, single nucleobase changes in the core of the predicted E2F binding site (+22 C-to-G or +23 G-to-C) caused full de-repression of the *A3B-luciferase* reporter that could not be further increased by tTAg (*Figure 2E*). Single and combinatorial base substitution mutations in the CHR element also resulted in partial or full de-repression of the *A3B-luciferase* reporter (*Figure 2F*). For instance, mutation of +10 TC-to-CT or +9 TT-to-AA caused partial reporter de-repression, which could still be further enhanced by tTAg. In contrast, mutating the two adenine nucleobases at the 3' end of the CHR element (+13 AA-to-TT) resulted in full reporter de-repression which could not be increased by tTAg. These fine-mapping results showed that both the putative E2F binding site and the adjacent CHR element are essential for repressing *A3B* transcription.

## Targeted mutagenesis demonstrates a repressive role for the +21 to +28 E2F element in regulating endogenous *A3B* transcription independent of activation by the PKC/ncNF-κB pathway

The abovementioned work indicated recruitment of a repressive complex to a putative E2F binding site in the *A3B-luciferase* reporter, which was necessarily episomal and may not be subject to the same regulatory mechanisms as the chromosomal *A3B* gene. To directly ask whether the endogenous +21 to +28 E2F site is involved in *A3B* repression, CRISPR/Cas9 technology was used to disrupt this region in diploid MCF10A cells. Four independent targeted clones showed elevated A3B protein levels in comparison to control *lacZ* clones, consistent with a repressive function for the putative E2F binding site (*Figure 3—figure supplement 1A*). DNA sequencing revealed allelic differences between the four clones, which could explain at least part of the variability observed in A3B elevation (*Figure 3—figure supplement 1B*).

To confirm and extend these results, homology-directed repair (HDR) was used to introduce precise base substitution mutations into the +21 to +28 E2F site in the endogenous *A3B* promoter of an MCF10A derivative engineered to be hemizygous for the entire *A3B* gene (Materials and methods). Tandem base substitution mutations, C22G and G25C, were chosen to disrupt the E2F site and simultaneously preserve the locus by maintaining the overall G:C content and spatial relationships between promoter elements (*Figure 3A*). Seven independent clones were obtained with the desired two base substitution mutations (*Figure 3B*). All seven showed robust increases in both A3B protein and mRNA levels with differences potentially due to clonal variation (*Figure 3C–E*). The mRNA levels of related *A3* family members were unaffected, which further confirmed specificity of the targeted genomic changes (*Figure 3—figure supplement 1*). Immunoblots were also performed for RAD51, an established RB/E2F-target (*Dean et al., 2012*; *Müller et al., 2017*), to show that global E2F regulation is unperturbed (*Figure 3C–D*). These results demonstrated that the endogenous E2F site at base pairs +21 to +28 of the *A3B* promoter contributes to transcriptional repression in MCF10A cells.

To determine whether this *cis*-element is solely responsible for endogenous *A3B* upregulation by tTAg or whether multiple tTAg-responsive mechanisms may combine to exert the observed phenotype, tTAg was expressed in two representative HDR targeted MCF10A clones and two *lacZ* controls and A3B levels were analyzed by RT-qPCR and immunoblotting. Expression of tTAg resulted in two- to three-fold higher A3B levels in *lacZ* control clones (*Figure 3F*), similar to results above with the

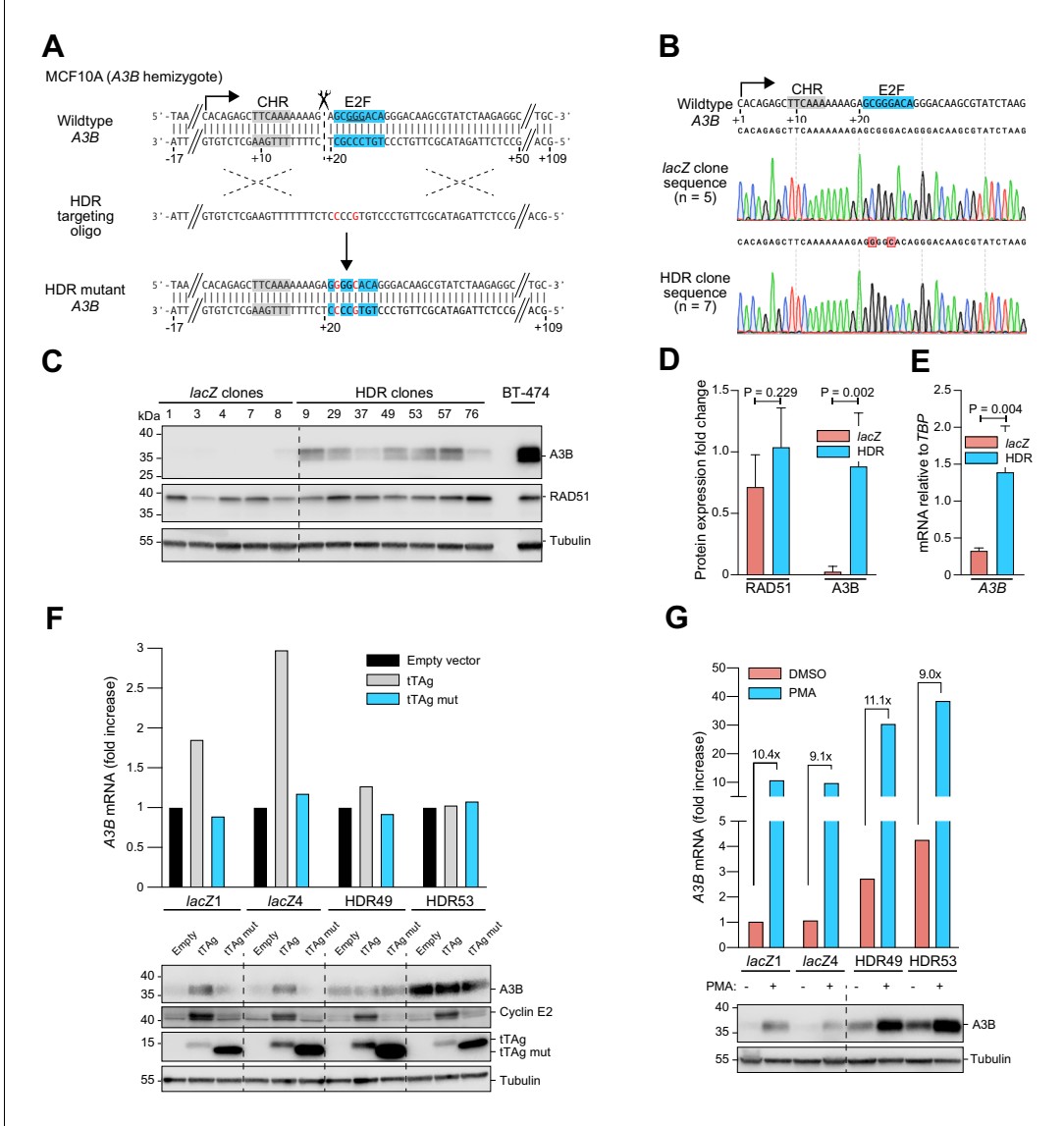

**Figure 3.** Single-base substitutions in the endogenous predicted E2F binding site induce *A3B* expression independent of activation by the PKC/ncNF-κ B pathway. Complementary supporting data are in *Figure 3—figure supplement 1*. (**A**) Schematic of CRISPR/Cas9-mediated HDR of the predicted E2F binding site in *A3B* hemizygous MCF10A cells. Top: CRISPR/Cas9 (scissors) introduces a DNA break (dashed line) adjacent to the predicted E2F binding site (blue). Middle: The ssDNA oligo used for HDR has two point mutations in the predicted E2F binding site including one that disrupts the PAM (underlined). Bottom: *A3B* promoter sequence of properly targeted clones. (**B**) Sanger DNA sequencing chromatograms of the E2F promoter region of a representative control clone (*lacZ* clones, n = 5) and a representative clone with the targeted E2F point mutations (HDR clones, n = 7). (**C–D**) A3B and RAD51 protein levels in control *lacZ* and HDR clones with tubulin as a loading control (representative immunoblots and quantification from n = 3 experiments). A3B-overexpressing BT-474 cells were used as a positive control. P-values from unpaired t-test. (**E**) *A3B* mRNA expression levels in control *lacZ* and HDR clones quantified by RT-qPCR (mean ± SD; p-value from unpaired t-test). (**F**) RT-qPCR (top) and immunoblot (bottom) results showing the effects of wildtype and LxCxE mutant tTAg on the *A3B* gene (top) and protein (bottom) expression in two representative *lacZ* and HDR49 clones. Cyclin E2 was used as a positive immunoblot control for tTAg-mediated induction of an RB/E2F-repressed gene. Tubulin was used as an immunoblot loading control. (**G**) Expression of A3B mRNA (top) and protein (bottom) upon PMA-treatment of the indicated *lacZ* control and HDR mutant clones. The magnitude of mRNA induction is indicated for each DMSO control and PMA-treated pair. Tubulin was used as an immunoblot loading control.

The online version of this article includes the following figure supplement(s) for figure 3:

**Figure supplement 1.** CRISPR/Cas9 disruption of the endogenous E2F site in the *A3B* promoter.

episomal reporter. In contrast, neither expression of an LxCxE mutant nor an empty mCherry control vector induced A3B. Importantly, tTAg had no effect on A3B mRNA or protein levels in the HDR targeted MCF10A clones (*Figure 3F*). This result was clear despite the fact that the LxCxE mutant was expressed more strongly than wildtype tTAg (likely due to loss of an autoregulatory mechanism yet-to-be-defined) and that some variability in endogenous A3B expression was observed from experiment-to-experiment (even using the same HDR-targeted clone). Nevertheless, these results combined to demonstrate that all of the observed A3B induction by tTAg is mediated by this single endogenous E2F site.

In parallel, representative HDR-targeted clones and *lacZ* controls were used to ask how this endogenous E2F site might impact A3B induction by PMA through the PKC/ncNF-κB signal transduction pathway (*Leonard et al., 2015*). This was done by treating cells with PMA and then quantifying A3B levels by RT-qPCR and immunoblotting. Interestingly, PMA caused similar induction of A3B mRNA and protein levels from both the wildtype endogenous promoter (*lacZ* controls) as well as the HDR-engineered promoter with tandem base substitution mutations C22G and G25C (*Figure 3G*). Overall, simultaneous de-repression through HDR-targeted mutation of the single E2F site and activation by PMA caused a thirty-fold increase in *A3B* levels above the uninduced basal level in the *lacZ* controls. Together with the above data above, these results demonstrated that *A3B* expression is impacted independently by tTAg/E2F and PKC/ncNF-κB and signal transduction mechanisms.

## Repressive E2F4/DREAM and E2F6/PRC1.6 complexes bind to the *A3B* promoter

Collectively, the data so far indicate that the putative E2F binding site is functionally relevant in repressing both *A3B-luciferase* reporter activity and endogenous *A3B* expression. However, the identity of the repressive complex(es) bound to this *cis*-element was unclear because multiple E2F family members are capable of transcriptional repression (Introduction). To address this problem in an unbiased manner, a series of proteomic experiments was conducted to identify MCF7 nuclear proteins capable of binding to the wildtype *A3B* +1 to +50 promoter sequence but not to repression-defective mutants (see *Figure 4A* for a schematic of the proteomics workflow and Materials and methods for details). This approach was facilitated by stable isotope labeling with amino acids in cell culture (SILAC) to create heavy (H) and light (L) nuclear extracts for H versus L and L versus H comparisons with the different promoter substrates. Interestingly, in an experiment comparing proteins bound to the wildtype *A3B* promoter sequence versus a promoter sequence with mutations spanning the predicted E2F binding site (matching the +22-to-30 mutant in *Figure 1D–E*), a greater than four-fold enrichment was observed for almost all proteins in the repressive DREAM complex, including TFDP1, TFDP2, RBL1, RBL2, E2F4, E2F5, and the MuvB components LIN9, LIN37, LIN52, and LIN54 (*Figure 4B–C* and *Supplementary file 2*; confirmatory immunoblots for representative enriched proteins in *Figure 4—figure supplement 1*). Given that a single-base substitution +22 C-to-G was sufficient for full de-repression in reporter assays (*Figure 2E*), we repeated the SILAC DNA pull-downs comparing the wildtype promoter sequence and this mutant. Importantly, again, most members of the DREAM complex preferentially bound to the wildtype but not to the *A3B* mutant promoter sequence (*Figure 4B,D*, *Supplementary file 2*). Similar enrichments for DREAM complex components were also evident in a separate proteomics experiment comparing MCF7 nuclear proteins bound to the wildtype *A3B* promoter versus a promoter sequence with mutations spanning the CHR element (matching the +12-to-20 mutant in *Figure 1D–E*; *Figure 4B, E*, *Supplementary file 2*). These additional results indicated that the CHR site is also required for *A3B* promoter binding by the DREAM complex and that the E2F site alone is insufficient.

Interestingly, the proteomics data sets also implicated components of the PRC1.6 complex in binding to wildtype but not to E2F or CHR mutant *A3B* promoter sequences. In particular, E2F6, MGA, and L3MTBL2 were found enriched repeatedly (*Figure 4B–E*, *Supplementary file 2*, and confirmatory immunoblots for representative enriched proteins in *Figure 4—figure supplement 1*). Two additional PRC1.6 components, PCGF6 and WDR5, also approached the four-fold cut-off in one dataset (*Supplementary file 2*). These results indicated that the repressive PRC1.6 complex is also capable of binding to the wildtype *A3B* promoter sequence and may therefore also play a role in suppressing expression.

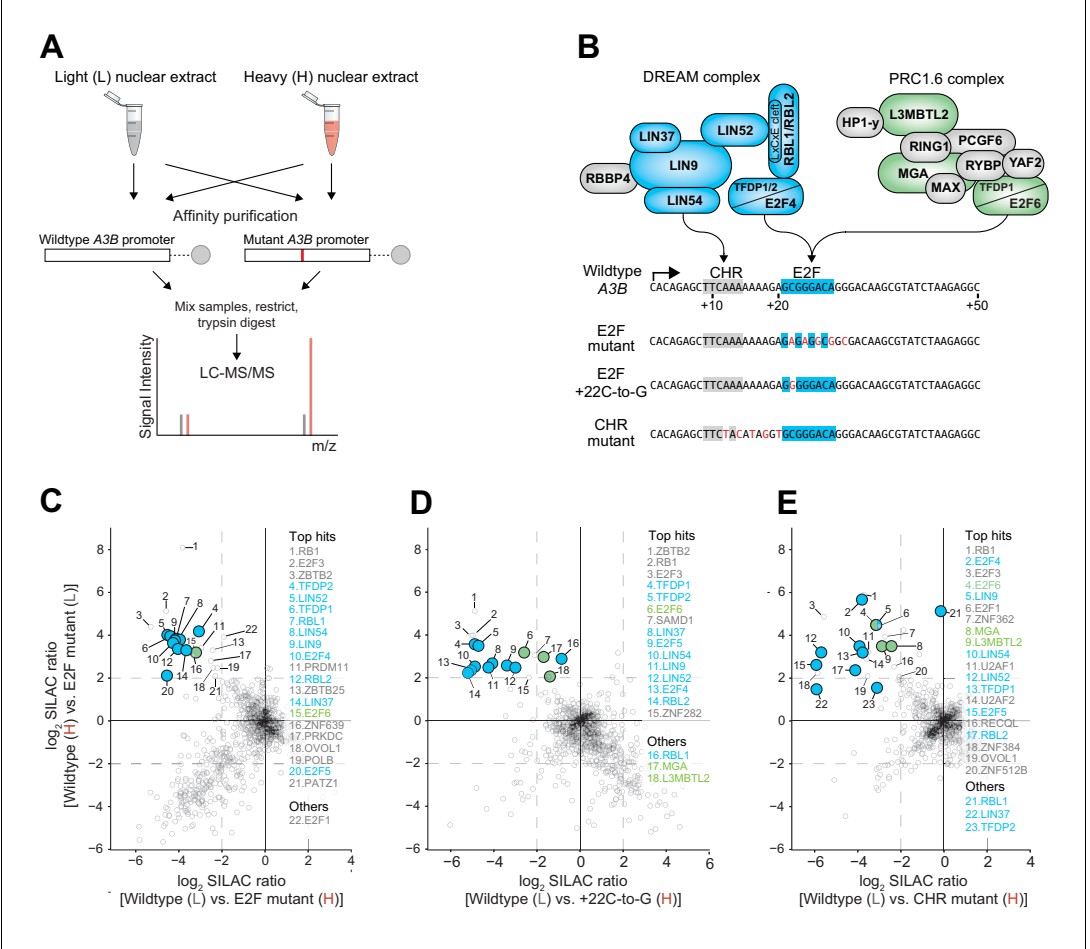

**Figure 4.** The DREAM and PRC1.6 repressive complexes bind to the CHR-E2F region of the *A3B* promoter. Immunoblot validations of representative binding proteins from proteomic experiments are in *Figure 4—figure supplement 1*. (**A**) Schematic of the SILAC DNA pull-down strategy used to identify proteins from MCF7 cells capable of interacting with *A3B* promoter sequences. (**B**) Illustration of DREAM and PRC1.6 complexes positioned over the indicated *A3B* promoter elements (proteomic hits shaded blue and green, respectively). (**C–E**) Log$_2$-transformed SILAC ratios of proteins purified using the indicated promoter sequences and identified through LC-MS/MS (dashed line, SILAC ratio threshold >2.0 [log$_2$]). 'Top hits' are proteins surpassing the >2.0 log$_2$ SILAC ratio threshold in both datasets (rank based on heavy versus light SILAC ratio). 'Others' are proteins of interest surpassing the >2.0 log$_2$ SILAC ratio threshold in at least one dataset. Data for DREAM and PRC1.6 components are shaded blue and green, respectively.

The online version of this article includes the following figure supplement(s) for figure 4:

**Figure supplement 1.** Immunoblot validations of representative *A3B* promoter-binding proteins identified in proteomics experiments.

## E2F4 and E2F6 complexes participate in *A3B* transcriptional repression

A series of chromatin immunoprecipitation (ChIP) experiments was done to determine whether *A3B* repression in non-tumorigenic MCF10A cells is mediated by one or both of the identified E2F complexes. Although prior work has implicated the E2F4/DREAM complex (*Periyasamy et al., 2017*), the potential involvement of E2F6/PRC1.6 is novel. Anti-E2F4 and anti-E2F6 antibodies were used to immunoprecipitate cross-linked transcriptional regulatory complexes from MCF10A *lacZ*4 (control) and HDR49 (E2F site E mutant) cells described above and promoter occupancy was determined by quantitative PCR (*Figure 5A–B*). The wildtype *A3B* promoter in *lacZ*4 cells showed similarly strong enrichment for binding by both E2F4 and E2F6, and single-base substitutions in E2F site E in HDR49 cells reduced binding of both proteins to background levels. In parallel analyses, significant E2F4 enrichment was evident in the promoter regions of two established E2F4/DREAM-repressed genes, *RAD51* and *TTK* (*Dean et al., 2012*; *Engeland, 2018*; *Müller et al., 2017*). E2F6 was enriched only at the *RAD51* promoter and not the *TTK* promoter.

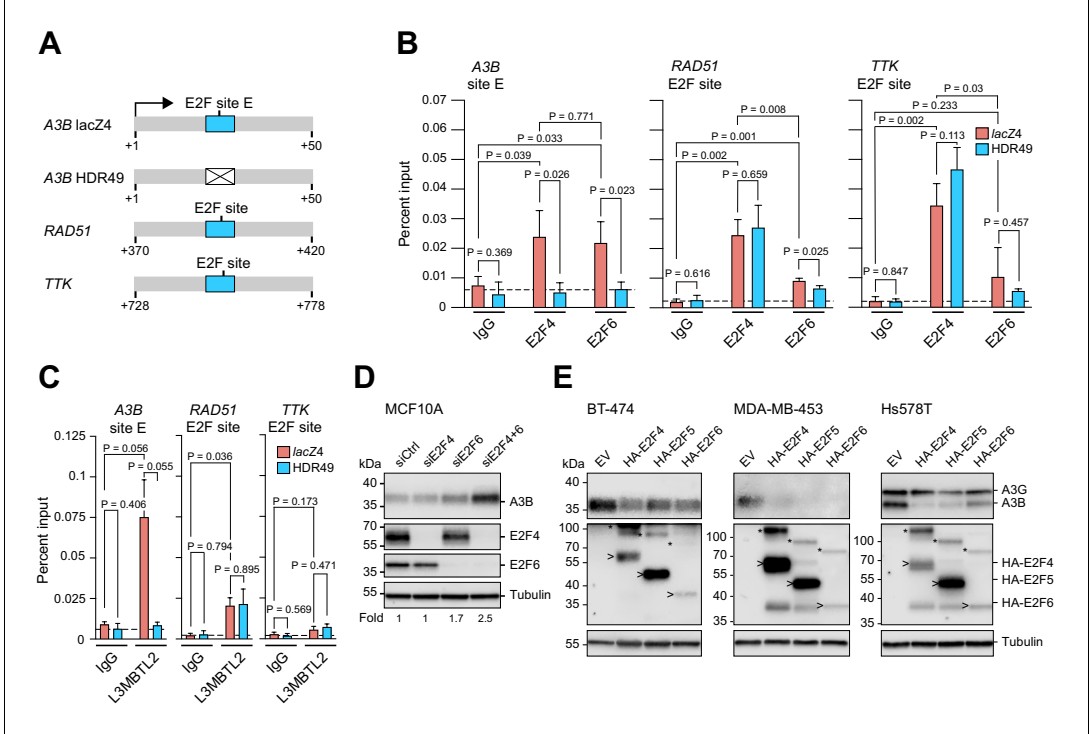

**Figure 5.** Endogenous *A3B* regulation by both E2F4/DREAM and E2F6/PRC1.6 complexes. (**A**) Schematics of the promoter regions interrogated by ChIP experiments. Wildtype E2F sites are depicted by blue boxes and the mutant E2F site in the A3B promoter by a white X-box. (**B–C**) E2F4, E2F6, and L3MBTL2 occupancy at the indicated E2F sites in *A3B*, *RAD51*, and *TTK*, as analyzed by ChIP-qPCR using *lacZ*4 and HDR49 cells. Experiments in (**B**) report mean ± SD of n = 3 biologically independent replicates and in (**C**) of n = 2 biologically independent replicates (p values from unpaired t-test). Dashed lines indicate the average IgG background. (**D**) Immunoblots of A3B, E2F4, and E2F6 in MCF10A cells treated 24 hr with the indicated siRNAs. Tubulin was used as a loading control. Representative blots are shown and fold-changes below are based on the average values from n = 3 biologically independent replicates. (**E**) Immunoblots of A3B and the indicated HA-tagged E2F proteins in BT-474, MDA-MB-453, and Hs578T cells transduced with lentiviral constructs encoding mCherry-T2A-HA-E2F4, -HA-E2F5, -HA-E2F6, or -EV (empty vector control). Representative blots of n = 2 experiments are shown. Asterisks indicate E2Fs still fused with mCherry due to incomplete ribosome skipping at the T2A site, and arrowheads indicate bands for free E2F. Tubulin was used as a loading control.

The higher E2F6 signal at the *A3B* promoter compared to the *RAD51* and *TTK* promoters prompted us to ask whether other PRC1.6 components may also bind preferentially. ChIP experiments for L3MBTL2 revealed strong binding of this PRC1.6 component to the *A3B* promoter, intermediate levels to the *RAD51* promoter, and insignificant levels to the *TTK* promoter (*Figure 5C*). These ChIP experiments indicated that the *A3B* promoter can be occupied by both E2F4 and E2F6 complexes, that the binding of either complex requires an intact +21 to +28 E2F site, and that the binding of the same proteins to other established E2F sites can vary significantly within the same cell population.

Next, we used small interfering (si)RNAs to interrogate the repressive function of each complex in MCF10A cells. Surprisingly, E2F4 depletion alone did not alter A3B expression, whereas E2F6 depletion caused an increase in A3B protein levels by immunoblotting (*Figure 5D*). We also observed that combined E2F4/E2F6 depletion increases A3B protein levels more than E2F6 alone, indicating that both complexes contribute to repression with the latter potentially being more dominant. Analogous knockdown attempts in MCF7 cells caused overt distress and inviability (data not shown). Conversely, overexpression of either E2F4 or E2F6, as well as E2F5 which also forms a DREAM complex (*Litovchick et al., 2007*), was able to repress A3B expression to varying extents in multiple different breast cancer cell lines (*Figure 5E*). Taken together, the ChIP, knockdown, and overexpression studies indicate that both E2F4/DREAM and E2F6/PRC1.6 complexes can occupy the *A3B* promoter and repress transcription. Moreover, the significant A3B upregulation observed upon E2F6 knockdown

but not E2F4 knockdown suggests that the PRC1.6 complex repression mechanism may predominate.

## Breast tumors with overexpression of an E2F-repressed gene set elicit higher levels of APOBEC signature mutations

Breast tumors frequently display *A3B* overexpression and APOBEC signature mutations (*Alexandrov et al., 2013*; *Angus et al., 2019*; *Bertucci et al., 2019*; *Burns et al., 2013a*; *Burns et al., 2013b*; *Nik-Zainal et al., 2012*; *Nik-Zainal et al., 2016*; *Roberts et al., 2013*). However, association studies with large breast cancer cohorts have shown only weak positive or negligible associations between *A3B* expression levels and APOBEC signature mutation loads, and clear outliers exist including tumors with high *A3B* and few APOBEC signature mutations and low *A3B* and many APOBEC signature mutations (*Buisson et al., 2019*; *Burns et al., 2013a*; *Burns et al., 2013b*; *Nik-Zainal et al., 2014*; *Roberts et al., 2013*). This variability may be due to a number of factors including different durations of mutagenesis (i.e. tumor age is unknown and distinct from a patient's biological age) and mutagenic contributions from other APOBEC3 enzymes governed by distinct regulatory mechanisms (*Buisson et al., 2019*; *Cortez et al., 2019*; *Nik-Zainal et al., 2014*; *Starrett et al., 2016*). However, given our results implicating both E2F4 and E2F6 complexes in *A3B* repression, we reasoned that effects from these and other potentially confounding variables may be overcome by asking whether the APOBEC mutation signature is enriched in breast tumors with functional overexpression of an E2F-repressed 20 gene set.

This was done by analyzing TCGA breast cancer RNA-seq and whole-exome sequencing data (*Cancer Genome Atlas Network, 2012*) for gene expression levels and base substitution mutation signatures (workflow in *Figure 6A*). The top 20 genes associating positively with *A3B* and also showing evidence for E2F repression (*Litovchick et al., 2007*; *Müller et al., 2014*) were used to rank tumors based on highest to lowest expression levels of each gene (*Figure 2A–B* and *Supplementary files 1* and *3*). Tumors ranking in the top or bottom quartiles for expression of all 20 genes were considered for additional analyses (n = 53 and n = 111 tumors in the common high and common low groups, respectively). Once common high and low groups were delineated, pairwise comparisons were made for *A3B* expression levels, percentage of APOBEC signature mutations, and APOBEC signature enrichment values. As expected from the analysis work-flow and the likelihood of a shared transcriptional regulation mechanism, tumors with common high-expressing genes showed an average of twenty-fold higher *A3B* mRNA levels than tumors with common low-expressing genes (p<2.4×10$^{-6}$ by Welch's test; *Figure 6B*). More interestingly, tumors with the common high-expressing genes showed an average of 9.3% APOBEC signature mutations versus 3.2% in the common low group (p=0.026 by Welch's test; *Figure 6C*). As an independent metric, significantly higher APOBEC mutation signature enrichment values were evident in tumors defined by the common set of high-expressed genes in comparison to tumors with the same genes expressed at low levels (p=0.003 by Welch's test; *Figure 6D*). A pairwise analysis of the mean expression value of the top 20 *A3B*-associating/E2F-repressed genes yielded similar positive associations with *A3B* mRNA expression levels, APOBEC mutation percentages, and enrichment scores (*Figure 6—figure supplement 1*).

Although associations between *A3B* mRNA levels and APOBEC mutation signature have been analyzed previously (references above), we wanted to apply an *A3B*-focused quartile-binning approach to be able to compare results with those from the 20-gene set above (work-flow in *Figure 6E*). Therefore, TCGA breast tumor RNA-seq data were used to identify the top 25% and bottom 25% of *A3B* expressing tumors (n = 179 per group). As mentioned above and expected from the work-flow, average *A3B* mRNA levels were much higher in the *A3B*-high group in comparison to the *A3B*-low group (*Figure 6F*). Also similar to the analysis above, both the average APOBEC mutation signature percentages and average APOBEC enrichment scores trended upward in A3B-high tumors (*Figure 6G–H*). However, in contrast to the analysis above, the difference in APOBEC mutation signature percentages was not significant and the difference in APOBEC enrichment scores was barely significant (p=0.154 and 0.042 by Welch's test, respectively). Altogether, these results indicate that coordinated overexpression of an RB/E2F-repressed gene set may be a better indicator for APOBEC mutation susceptibility than expression of *A3B* itself. Potential explanations for the different results from each analysis approach are discussed below.

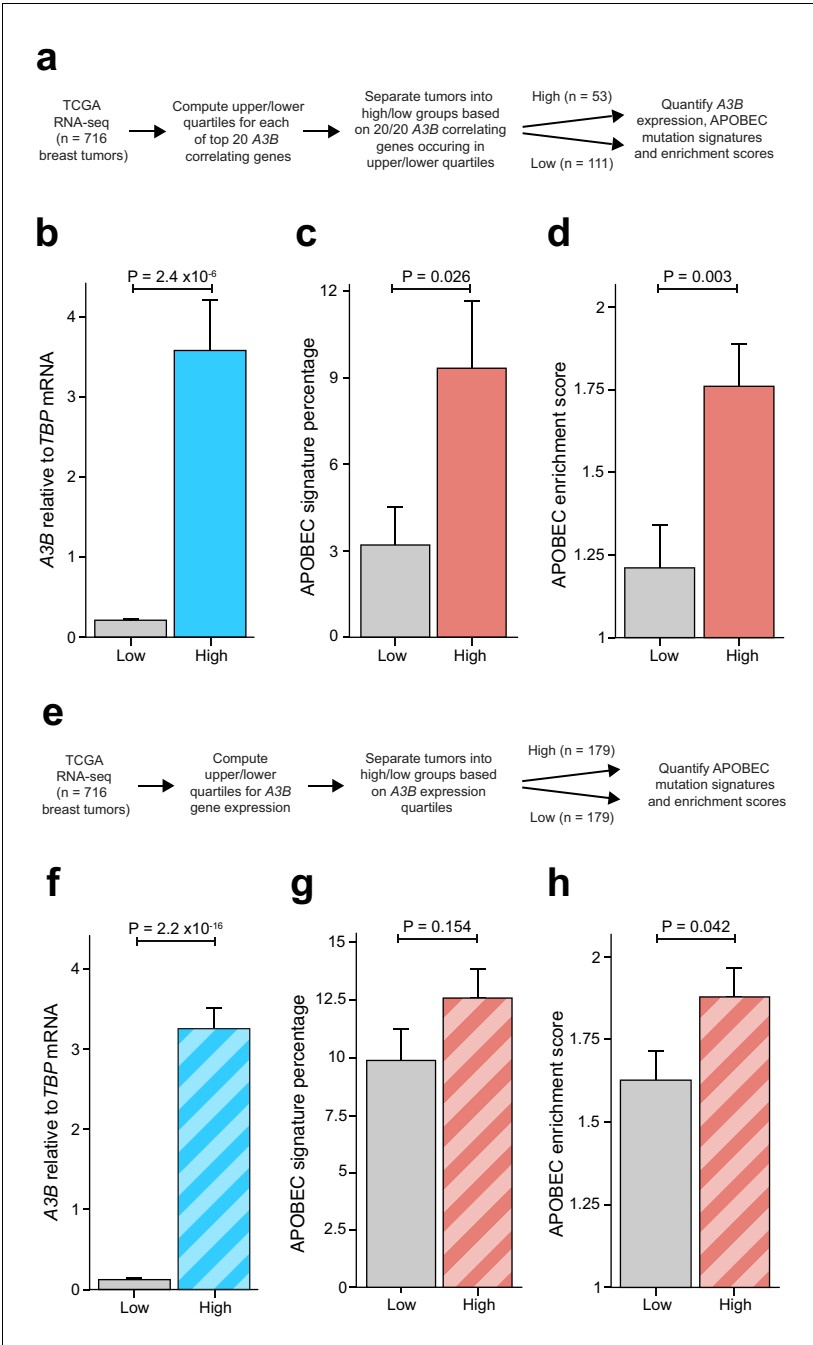

**Figure 6.** Elevated levels of APOBEC signature mutations in breast tumors with coordinated overexpression of an E2F-repressed gene set. Complementary analyses are presented in *Figure 6—figure supplement 1*. (A) Schematic depicting the bioinformatics workflow of TCGA breast tumor data sets based on the 20 genes most strongly associated with *A3B* expression (*Figure 2A* and *Supplementary file 1*). (B–D) The mean *A3B* mRNA levels, mean APOBEC mutation percentages, and mean APOBEC enrichment scores in breast tumors with coordinated overexpression (high) or repression (low) of the 20 gene set (mean ± SD; n = 53 tumors in the high group and n = 111 in the low group; p values from Welch's t-test). (E) Schematic depicting the bioinformatics workflow of TCGA breast tumor data sets based solely on *A3B* mRNA expression levels. (F–H) The mean *A3B* mRNA levels, mean APOBEC mutation percentages, and mean APOBEC enrichment scores in breast tumors with high or low *A3B* mRNA levels (mean ± SD of top and bottom quartiles; n = 179 tumors in each group; p values from Welch's t-test).

The online version of this article includes the following figure supplement(s) for figure 6:

*Figure 6 continued on next page*

*Figure 6 continued*

**Figure supplement 1.** Global pairwise comparisons of the mean mRNA levels of the top 20 E2F-repressed/*A3B*-associated genes and *A3B* mRNA levels and APOBEC mutation signature prevalence in primary breast tumors.

## Discussion

The studies here are the first to demonstrate that two repressive E2F complexes, E2F4/DREAM and E2F6/PRC1.6, combine to suppress *A3B* transcription and thereby protect genomic integrity in normal cells. The construction of a novel *A3B-luciferase* reporter enabled the delineation of a repressive *cis*-element comprised of juxtaposed E2F and CHR sites. Site-directed mutation of either site caused full de-repression that could not be further enhanced by co-expression of BK-PyV tTAg. These results indicated that TAg-mediated upregulation of *A3B* reported previously (*Starrett et al., 2019*; *Verhalen et al., 2016*) is occurring exclusively through the RB/E2F axis and not through an alternative LxCxE-dependent mechanism. The importance of this E2F binding site in the endogenous *A3B* promoter was demonstrated definitively by CRISPR/Cas9-mediated base substitution mutation and experimentation with a panel of independent knock-in clones. Proteomics experiments revealed that two distinct repressive regulatory complexes, specifically E2F4/DREAM and E2F6/PRC1.6, are capable of binding to the wildtype *A3B* promoter but not to E2F or CHR mutant derivatives. Repressive roles for both E2F complexes were demonstrated by ChIP, knockdown, and overexpression studies. Finally, the potential pathological significance of E2F-mediated de-repression of *A3B* in breast cancer was supported by TCGA data analyses showing significant positive associations between elevated expression of a set of 20 coordinately expressed E2F-regulated genes and higher levels of APOBEC signature mutations.

There is a broad interest in understanding the molecular mechanisms that govern *A3B* transcriptional regulation due to its physiological functions in antiviral immunity and pathological roles in cancer mutagenesis. Although prior studies implicated the E2F4/DREAM complex and generally the RB/E2F axis in repressing *A3B* transcription (*Periyasamy et al., 2017*; *Starrett et al., 2019*), the work here is the first to define the responsible *cis*-elements (juxtaposed CHR and E2F sites), show that all PyV tTAg-mediated activation occurs through this single bipartite sequence, and demonstrate coordinated repression not only by the E2F4/DREAM complex but, surprisingly, also by the E2F6/PRC1.6 complex. Moreover, *A3B* induction by E2F4/6 de-repression occurs independently of *A3B* activation by PKC/ncNF-κB signal transduction. This additional result suggests that upregulation of A3B expression through genetic or viral perturbation of the RB/E2F cell cycle pathway has the potential to combine synergistically with inflammatory responses and trigger even greater levels of genomic DNA damage and mutagenesis. The role of p53 in *A3B* transcriptional regulation is less clear with some studies indicating that p53 inactivation leads to *A3B* upregulation (*Menendez et al., 2017*; *Periyasamy et al., 2017*) and others demonstrating that *TP53* knockout has no effect on *A3B* transcription (*Nikkilä et al., 2017*; *Starrett et al., 2019*). This may be due to differences in cell types and growth conditions. Alternatively, rather than playing an upstream role in *A3B* transcriptional regulation, p53 may function to help activate a downstream DNA damage response to prevent the accumulation of mutations by A3B, which also explains why genetic inactivation of *TP53* associates positively with elevated *A3B* mRNA levels (*Burns et al., 2013a*).

Our results support a model in which E2F4/DREAM and E2F6/PRC1.6 complexes combine to repress A3B transcription (*Figure 7*). These two complexes are likely to compete for binding to the same conserved E2F site located at +21 to +28 of the *A3B* promoter because tandem base substitution mutations (C22G and G25C) de-repress expression of endogenous *A3B* and render the locus non-responsive to further activation by tTAg (*Figure 3F*). Similar results were obtained using E2F site E mutants of the episomal *A3B-luciferase* reporter (*Figure 2E*). Base substitution mutations in the adjacent CHR site in the episomal *A3B-luciferase* reporter also caused *A3B* de-repression to levels that could not be further increased by tTAg (*Figure 2E*). These genetic results were corroborated by proteomics data sets indicating that base substitution mutations in either the E2F site or the CHR site fully abrogate promoter sequence binding by both the DREAM and PRC1.6 complexes (*Figure 4*). However, unlike E2F4, E2F6 is not known to be regulated through a TAg/LxCxE-dependent mechanism nor has its function been shown to require a CHR site. Future work will be required to bridge this knowledge gap. For instance, it may be possible that a subset of E2F6/PRC1.6 complex

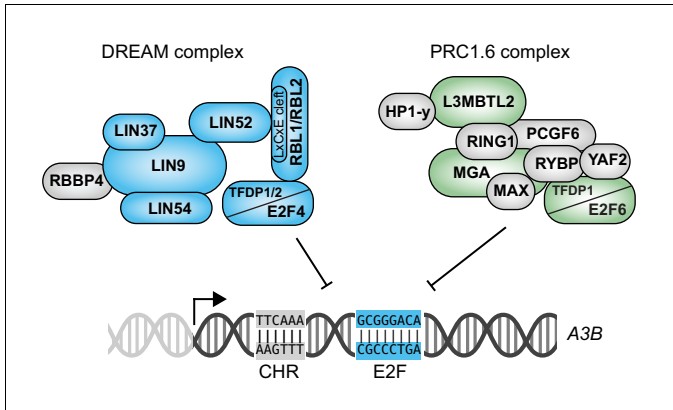

**Figure 7.** Model for coordinated repression of *A3B* transcription by both E2F4/DREAM and E2F6/PRC1.6 complexes. Transcriptional repression of *A3B* through the combined activities of E2F4/DREAM and E2F6/PRC1.6 complexes. Other regulatory mechanisms including *A3B* transcriptional activation by PKC/ncNF-κB signal transduction are not shown. See text for details and discussion.

leverages an as-yet-unknown CHR binding factor to repress genes such as *A3B*. Alternatively, it may be possible that LxCxE-dependent interactions with PRC1.6 components other than E2F6 might interfere with the repressive function of PRC1.6. It is unlikely, however, that the E2F6/PRC1.6 complex requires the E2F4/DREAM complex as a cofactor for binding because E2F4-depleted cells maintain near-complete repression of *A3B* expression (*Figure 5D*).

The E2F-governed regulatory mechanism described here provides an attractive explanation for a large proportion of reported *A3B* overexpression in both viral and non-viral cancer types. For instance, the HPV E7 and PyV TAg oncoproteins may trigger *A3B* upregulation directly by dissociating repressive E2F complexes. Accordingly, cervical cancers are almost invariably HPV-positive, *A3B*-overexpressing, and enriched for APOBEC signature mutations (*Burns et al., 2013b*; *Cancer Genome Atlas Research Network, 2017*; *Roberts et al., 2013*; *Zapatka et al., 2020*). HPV-positive head/neck cancers also show *A3B*-overexpression and APOBEC mutation signature enrichment (*Burns et al., 2013b*; *Cancer Genome Atlas Network, 2015*; *Cannataro et al., 2019*; *Faden et al., 2017*; *Roberts et al., 2013*; *Vieira et al., 2014*; *Zapatka et al., 2020*). Importantly, many HPV-negative cancers elicit similarly high *A3B* expression levels and APOBEC mutation burdens (*Burns et al., 2013b*; *Cancer Genome Atlas Network, 2015*; *Cannataro et al., 2019*; *Gillison et al., 2019*). Moreover, HPV status in head/neck cancer appears mutually exclusive with alterations of RB/E2F axis genes, such that HPV-negative cancers often display copy number loss of *CDKN2A* (encoding p16) and overexpression of *CCND1* (encoding Cyclin D1; *Cancer Genome Atlas Network, 2015*; *Gillison et al., 2019*; *Zapatka et al., 2020*), which effectively mimics a subset of the oncogenic effects of E7. This indicates that both virus-dependent and independent tumors may exploit the same pathway to derepress *A3B* and gain an evolutionary advantage. This possibility is also supported by frequent lesions in the RB/E2F pathway in breast cancer, including loss of *RB1*, *CDKN1B* (encoding p27), and *CDKN2A* as well as amplification of *CCND1* (*Angus et al., 2019*; *Bertucci et al., 2019*; *Cancer Genome Atlas Network, 2015*; *Ertel et al., 2010*; *Nik-Zainal et al., 2016*; *Cancer Genome Atlas Network, 2012*).

Our studies also raise the possibility that high levels of expression of a set of 20 normally E2F-repressed genes may be used to identify tumors with elevated levels of APOBEC signature mutations (*Figure 6A–D* and *Figure 6—figure supplement 1*). Such information could be useful, for instance, to help identify the subset of patients with hypermutated tumors that may be most responsive to immunotherapy. It is also interesting that *A3B* mRNA levels do not associate as strongly with APOBEC signature mutation loads or enrichment values (*Figure 6E–H*). This discordance is unexpected and may be due to a combination of factors including cell cycle dysregulation (magnitude and mechanism), DNA damage response and DNA repair capabilities (including p53 functionality), tumor microenvironment (including inflammation and infection status), and possible contributions from related APOBEC3 family members including A3A and A3H. For instance, a more rapid cell

cycle, dampened or disabled cell cycle checkpoints, downregulated (or saturated) DNA repair mechanisms, and potential coordination with A3A (*Figure 2A*) may combine to create conditions favoring an overall accumulation of APOBEC signature mutations. The overall APOBEC signature may also be influenced by an *A3A-B* fusion allele but its low frequency in TCGA data sets precluded analysis here. We are particularly intrigued by the potential for synergistic *A3B* induction by simultaneous E2F de-repression as part of cell cycle dysregulation and inflammation (modeled here by PyV tTAg expression and PMA treatment, respectively, in *Figure 3F–G*). These perturbations, especially in combination with others such as viral or mutational inactivation of p53, may both activate the APOBEC mutation program and create an optimal environment for DNA damage tolerance, mutation accumulation, and tumor evolution.

# Materials and methods

## Key resources table

| Reagent type (species) or resource | Designation | Source or reference | Identifiers | Additional information |
|---|---|---|---|---|
| Cell line (*Homo sapiens*, female) | MCF10A | ATCC | Cat#:CRL-10317 RRID:CVCL_0598 | |
| Cell line (*Homo sapiens*, female) | MCF10A-4C10 | This study | Hemizygous for *A3B* | Request by contacting RSH |
| Cell line (*Homo sapiens*, female) | MCF7 | ATCC | Cat#:HTB-22 RRID:CVCL_0031 | |
| Cell line (*Homo sapiens*, female) | BT474 | ATCC | Cat#:CLR-7913 RRID:CVCL_0179 | |
| Cell line (*Homo sapiens*, female) | Hs578T | ATCC | Cat#:HTB-126 RRID:CVCL_0332 | |
| Cell line (*Homo sapiens*, female) | MDA-MB-453 | ATCC | Cat#:HTB-131 RRID:CVCL_0418 | |
| Cell line (*Homo sapiens*, female) | HEK 293T | ATCC | Cat#:CRL-3216 RRID:CVCL_0063 | |
| Antibody | Anti-RAD51 (rabbit monoclonal) | Abcam | Cat#:ab133534 RRID:AB_2722613 | WB (1:10,000) |
| Antibody | Anti-E2F4 (mouse monoclonal) | Santa Cruz | Cat#:sc-398543 | WB (1:250) ChIP: 5 µg per 20 µg Dynabeads |
| Antibody | Anti-E2F6 (rabbit polyclonal) | Abcam | Cat#:ab53061 RRID:AB_2097254 | WB (1:500) ChIP: 5 µg per 20 µg Dynabeads |
| Antibody | Anti-HA (rabbit monoclonal) | Cell Signaling | Cat#:3724 RRID:AB_1549585 | WB (1:5000) |
| Antibody | Anti-Rb (mouse monoclonal) | Santa Cruz | Cat#:sc-102 RRID:AB_628209 | WB (1:300) |
| Antibody | Anti-E2F1 (mouse monoclonal) | Santa Cruz | Cat#:sc-251 RRID:AB_627476 | WB (1:1,000) |
| Antibody | Anti-E2F3 (mouse monoclonal) | Santa Cruz | Cat#:sc-56665 RRID:AB_1122397 | WB (1:800) |
| Antibody | Anti-E2F5 (mouse monoclonal) | Santa Cruz | Cat#:sc-374268 RRID:AB_10988935 | WB (1:800) |
| Antibody | Anti-E2F6 (mouse monoclonal) | Santa Cruz | Cat#:sc-53273 RRID:AB_783163 | WB (1:300) |
| Antibody | Anti-LIN9 (mouse monoclonal) | Santa Cruz | Cat#:sc-398234 | WB (1:300) |
| Antibody | Anti-tubulin (mouse monoclonal) | Sigma-Aldrich | Cat#:T5168 RRID:AB_477579 | WB (1:20,000) |
| Antibody | Anti-A3B (rabbit monoclonal) | NIH AIDS Reagent Program | Cat#:12397 RRID:AB_2721202 | WB (1:1,000) |

*Continued on next page*

*Continued*

| Reagent type (species) or resource | Designation | Source or reference | Identifiers | Additional information |
|---|---|---|---|---|
| Antibody | Anti-L3MBTL2 (rabbit polyclonal) | Active Motif | Cat#:39569 RRID:AB_2615062 | ChIP: 5 µg per 20 µg Dynabeads |
| Recombinant DNA reagent | pGL4.74 TK-RL renilla control (plasmid) | Promega | Cat#:E692A | Internal control for luciferase assays |
| Recombinant DNA reagent | pGL3 basic (plasmid) | Promega | Cat#:E1751 | Base vector for luciferase assays |
| Recombinant DNA reagent | pA3B-luciferase (plasmid) | This study | Wildtype *A3B* promoter + luciferase | Request by contacting RSH |
| Recombinant DNA reagent | pLenti-lox-empty vector (plasmid) | *Carpenter et al., 2019* | | |
| Recombinant DNA reagent | pLenti-lox-BKPyV tTAg (plasmid) | *Starrett et al., 2019* | | |
| Recombinant DNA reagent | pLenti-lox-BKPyV tTAg LxCxE mutant (plasmid) | *Starrett et al., 2019* | | |
| Recombinant DNA reagent | pLenti4/TO-mCherry-T2A-MCS (plasmid) | This study | Base vector for E2F expression | Request by contacting RSH |
| Recombinant DNA reagent | pLenti4/TO-mCherry-T2A-HA-E2F4 (plasmid) | This study | Lentiviral vector for expression of E2F4 | Request by contacting RSH |
| Recombinant DNA reagent | pLenti4/TO-mCherry-T2A-HA-E2F5 (plasmid) | This study | Lentiviral vector for expression of E2F5 | Request by contacting RSH |
| Recombinant DNA reagent | pLenti4/TO-mCherry-T2A-HA-E2F6 (plasmid) | This study | Lentiviral vector for expression of E2F6 | Request by contacting RSH |
| Recombinant DNA reagent | pLentiCRISPR-LoxP-A3B-gRNA#1 (plasmid) | This study | Lentiviral vector for expression of gRNA targeting E2F site E | Request by contacting RSH |
| Recombinant DNA reagent | pLentiCRISPR-LoxP-A3B-gRNA#3 (plasmid) | This study | Lentiviral vector for expression of gRNA targeting E2F site E | Request by contacting RSH |
| Sequence-based reagent | Cas9-encoding modified RNA | TriLink Biotech | Cat#:L7206-100 | |
| Commercial assay or kit | Dual Luciferase Reporter Assay | Promega | Cat#:E1960 | |
| Commercial assay or kit | Neon Transfection System 100 µL Kit | ThermoFisher | Cat#:MPK10025 | |
| Software, algorithm | MaxQuant version 1.5.2.8 | MaxQuant | RRID:SCR_014485 | |
| Software, algorithm | Fiji | Fiji | RRID:SCR_002285 | |
| Software, algorithm | GraphPad Prism 6 | GraphPad | RRID:SCR_002798 | |
| Software, algorithm | Image Studio | LI-COR Biosciences | RRID:SCR_015795 | |
| Other | Spark Multimode Microplate Reader | Tecan | | |
| Other | Neon Transfection System | ThermoFisher | Cat#:MPK5000 | |
| Other | LI-COR Odyssey FC | LI-COR | Cat#:2800 | |
| Other | LightCycler 480 Instrument | Roche | Cat#:04640268001 | |
| Other | EASY-nLC 1200 system | ThermoFisher | Cat#:LC140 | |
| Other | Q Exactive HF mass spectrometer | ThermoFisher | | |

## Cell lines and culture conditions

All cell lines were cultured at 37°C under 5% $CO_2$. MCF10A cells and derivative cell lines were grown in advanced DMEM/F-12 (Invitrogen) with HEPES and L-Glutamine, supplemented with 5% horse serum (Invitrogen), 20 ng/mL EGF (Peprotech), 0.5 mg/mL hydrocortisone (Sigma), 100 ng/mL cholera toxin (Sigma), 10 µg/mL recombinant human insulin (Sigma), penicillin (100 U/mL), and streptomycin (100 µg/mL). MCF7 cells were cultured as described (*Law et al., 2016*) for *A3B-luciferase* reporter assays and for proteomics in DMEM containing 10% dialyzed FBS (PAN-Biotech), penicillin (100 U/mL), and streptomycin (100 µg/mL). BT-474 and Hs578T cells were cultured in DMEM supplemented with 10% FBS (Invitrogen), penicillin (100 U/mL), and streptomycin (100 µg/mL). MDA-MB-453 and 293 T cells were cultured in RPMI supplemented with 10% FBS, penicillin (100 U/mL), and streptomycin (100 µg/mL). All cell lines tested negative for *Mycoplasma* using a PCR-based assay (*Uphoff and Drexler, 2011*). PMA (ThermoFisher) was used at 25 ng/mL for 6 hr.

## Plasmids and site-directed mutagenesis

The integrity of all plasmids was confirmed by Sanger sequencing. Oligos used for cloning, sequencing, and site-directed mutagenesis are listed in *Supplementary file 4*. The pLenti-lox constructs encoding BKPyV tTAg or the LxCxE mutant were described (*Starrett et al., 2019*). The *A3B* promoter sequence (−900 to +50 corresponding to chr22:39,377,504–39,378,453 of the GRCH37/hg19 assembly) was ordered as a gBlock (IDT), subjected to overhang extension PCR to add 5' KpnI and 3' NheI restriction sites, and then cut and ligated into compatibly digested pGL3-basic (Promega). Site-directed mutagenesis was done following standard procedures (Quickchange, Agilent).

E2F overexpression in BT-474 was done using pLent4/TO/V5-DEST (ThermoFisher), modified to lack the V5 tag through XhoI and AgeI digestion followed by insertion of a stuffer with compatible overhangs. 5' EcoRI and 3' AgeI sites were then added to a mCherry-T2A-MCS (multiple cloning site) cassette through overhang extension PCR and ligated into the base vector using compatible overhangs, resulting in the parental pLenti4/TO-mCherry-T2A-MCS vector. Then, coding regions of E2F4 (NM_001950.3), E2F5 (NM_001951.3 var 1), and E2F6 (NM_198256.3 var A) were cloned into pcDNA3.1, and a N-terminal HA-tag was inserted by site-directed mutagenesis. Finally, 5' NheI and 3' AgeI sites were added to the HA-tagged E2F sequences by overhang extension PCR, followed by ligation into compatibly digested pLenti4/TO-mCherry-T2A-MCS parental vector. Transduction of BT-474, MDA-MB-453, and Hs578T, plated at 300,000 cells per well of a six-well plate, was then performed with lentiviral particles produced in 293 T cells as described (*Burns et al., 2013a*; *Carpenter et al., 2019*; *Vieira et al., 2014*).

## Dual luciferase assays

MCF10A cells were plated at 50,000 cells per well and MCF7 at 100,000 cells per well in a 24-well plate, grown as described above, and transfected 24 hr later with a 1:2 ratio of plasmid cocktail and TransIT-2020 following vendor instructions (Mirus). Each transfection reaction was comprised of 250 ng luciferase reporter construct (pGL3-basic, pA3B-luciferase, or mutant derivatives), 10 ng pGL4.74 TK-RL renilla control plasmid, and 50 ng of pLenti-lox vector expressing BKPyV tTAg, tTAg LxCxE mutant, or empty control (*Starrett et al., 2019*). Lysates were prepared 48 hr later using the Dual Luciferase Reporter Assay according to manufacturer's instructions (Promega). Luminescence was detected using a Spark Multimode Microplate Reader (Tecan).

## CRISPR/Cas9-mediated editing of the *A3B* promoter

All sequences of oligos used during CRISPR/Cas9-mediated editing are listed in *Supplementary file 4*. pLenti-based CRISPR/Cas9 gene disruption was used initially to interrogate the *A3B* promoter using established protocols (*Carpenter et al., 2019*). Constructs targeting the *A3B* promoter or *lacZ* as a control were made using Golden Gate ligation and lentiviral particles were produced using 293 T cells (*Burns et al., 2013a*; *Carpenter et al., 2019*; *Vieira et al., 2014*). Transduction of 300,000 MCF10A cells per well of a six-well plate was followed 48 hr later by selection with puromycin. Individual clones were obtained by limited dilution and multi-week outgrowth. The promoter region from >6 clones per condition was amplified, cloned into pJET1.2 (ThermoFisher), and subjected to Sanger sequencing.

CRISPR/Cas9-mediated HDR was used to generate MCF10A clones with precise base substitutions in the *A3B* promoter. The MCF10A *A3B* hemizygous cell line was engineered by transducing MCF10A wildtype cells with pLentiCRISPR lentiviral particles expressing a single gRNA targeting the homologous 3'UTR of *A3A* and *A3B*, treating 48 hr with puromycin, and deriving single cell clones by limiting dilution. Clones were PCR-screened for alleles mimicking the natural *A3B* deletion (*Kidd et al., 2007*). A clone hemizygous for *A3B* was selected for precision editing of the +21 to +28 region of the *A3B* promoter. In short, 50,000 cells were transfected (Neon Transfection, Invitrogen) with 1 ng modified gRNA targeting the *A3B* promoter or *lacZ* (Synthego), 1.5 µg Cas9-encoding modified RNA (TriLink Biotech), and 6.25 pmol HDR targeting ssDNA oligo based on prior literature (*Prykhozhij et al., 2018*). The 5' and 3' terminal nucleotides of the ssDNA oligo were protected with phosphorothioates (*Richardson et al., 2016*). Clones were retrieved by limiting dilution 72 hr post transfection, outgrown for several weeks, and subjected to *A3B* promoter region DNA sequencing. Primers used for screening can be found in *Supplementary file 4*.

## Immunoblotting

For all immunoblot experiments, cells were harvested and counted using an automated cell counter (Countess, ThermoFisher). Pelleted cells were resuspended in PBS, and whole-cell protein extracts prepared by adding Laemmli reducing sample buffer followed by incubation at 98°C for 15 min. Protein expression was analyzed by immunoblot using standard laboratory techniques. Antibodies were rabbit anti-RAD51, 1:10,000 (Abcam, ab133534), mouse anti-E2F4, 1:250 (Santa Cruz, sc-398543), rabbit anti-E2F6, 1:500 (Abcam, ab53061), rabbit anti-HA, 1:5000 (Cell Signaling, C29F4), mouse anti-Rb, 1:300 (Santa Cruz, sc-102), mouse anti-E2F1, 1:1000 (Santa Cruz, sc-251), mouse anti-E2F3, 1:800 (Santa Cruz, sc-56665), mouse anti-E2F5, 1:800 (Santa Cruz, sc-374268), mouse anti-E2F6, 1:300 (Santa Cruz, sc-53273), mouse anti-LIN9, 1:300 (Santa Cruz, sc-398234) mouse anti-tubulin, 1:20,000 (Sigma-Aldrich, T5168), and rabbit anti-A3B, 1:1,000 [5210-87-13] (*Brown et al., 2019*).

mRNA quantification: mRNA was extracted (GenElute, Sigma-Aldrich) and cDNA was synthesized using SuperScript First-Strand RT (ThermoFisher). mRNA expression of all APOBEC3 family members and *TBP* was quantified by RT-qPCR with specific primers (*Refsland et al., 2010*) in Ssofast Supermix (Bio-Rad) using a Lightcycler (Roche). Primer sequences are listed in *Supplementary file 4*.

## ChIP experiments

ChIP experiments were done as described (*Leonard et al., 2015*) with minor modifications. A 15 cm plate with approximately $10^7$ sub-confluent cells was used as input for each immunoprecipitation. Chromatin was crosslinked for 10 min in 1% formaldehyde and then the crosslinking reaction was quenched using 125 mM glycine. Cells were washed in PBS, concentrated by centrifugation, and lysed in 1 mL Farnham lysis buffer (5 mM PIPES pH 8.0, 85 mM KCl, 1% Igepal CA-630, supplemented with protease inhibitors). After a 15 min incubation on ice, the cell nuclei were collected by 4°C centrifugation and then incubated 30 min on ice in nuclear lysis buffer (50 mM Tris-HCl pH8.1, 10 mM EDTA, 1% SDS, supplemented with protease inhibitors). Chromatin was sheared into 200–300 bp fragments using a Misonix sonicator for 13 cycles (30' on and 45' off) at an amplitude setting of 2. Chromatin was cleared of debris by centrifugation, diluted 5× with IP dilution buffer (50 mM Tris pH 7.4, 150 mM NaCl, 1% Igepal CA-630, 0.25% deoxycholic acid, 1 mM EDTA), and incubated with 5 µg of each mAb coupled to 20 µg Dynabeads Protein G magnetic beads (Invitrogen). Input controls of 1% were frozen down for later analysis. ChIP antibodies were mouse anti-E2F4 (Santa Cruz, sc-398543), rabbit anti-E2F6 (Abcam, ab53061), and rabbit anti-L3MBTL2 (Active Motif, 39569). After overnight incubation beads were washed twice with IP wash buffer 1 (50 mM Tris-HCl pH7.4, 150 mM NaCl, 1% Igepal CA-630, 0.25% deoxycholic acid, 1 mM EDTA), three times with IP wash buffer 2 (100 mM Tris-HCl pH 9.0, 500 mM LiCl, 1% Igepal CA-630) and once with IP wash buffer 3 (100 mM Tris-HCl pH 9.0, 500 mM LiCl, 1% Igepal CA-630, 1% deoxycholic acid, 150 mM NaCl). Chromatin was eluted in elution buffer (50 mM NaHCO$_3$1% SDS) for 30 min at 65°C, and reverse-crosslinked in an overnight reaction at 62°C in 500 mM NaCl, 50 mM EDTA, 100 mM Tris-HCl pH 6.8 and 2 mg proteinase K (Roche). DNA was cleaned up and concentrated using a ChIP DNA Clean and Concentrator kit (Zymo Research). Quantitative PCR reactions were done using specific primer sets (*Supplementary file 4*).

## RNAi-mediated knockdown

MCF10A cells were plated at 175,000 cells per well of a six-well plate and transfected the next day using 30 pmol siRNAs targeting E2F4 (SI02654694) and/or E2F6 (SI00375445; Qiagen) using the RNAiMAX protocol (Invitrogen). Samples were harvested 24 hr post transfection.

## SILAC labeling and DNA pull-down experiments

For SILAC labeling, MCF7 cells were incubated in DMEM (-Arg, -Lys) medium containing 10% dialyzed FBS (PAN-Biotech) supplemented with 42 mg/L $^{13}C_6^{15}N_4$L-arginine and 73 mg/L $^{13}C_6^{15}N_2$L-lysine (Cambridge Isotope) or the corresponding non-labeled amino acids, respectively. SILAC incorporation was verified by in-gel trypsin digestion and MS analysis of 'heavy' input samples to ensure an incorporation rate of >98%.

Cells were harvested and nuclear extracts were prepared as described (*Kappei et al., 2017*). Cells were harvested and incubated in hypotonic buffer (10 mM HEPES, pH 7.9, 1.5 mM MgCl₂, 10 mM KCl) on ice for 10 min. Cells were transferred to a Dounce homogenizer in hypotonic buffer supplemented with 0.1% Igepal CA630 (Sigma) and 0.5 mM DTT by 40 strokes. Nuclei were washed once in 1× PBS and extracted in hypertonic buffer (420 mM NaCl, 20 mM HEPES, pH 7.9, 20% glycerol, 2 mM MgCl₂, 0.2 mM EDTA, 0.1% Igepal CA630 (Sigma), 0.5 mM DTT) for 1 hr at 4°C on a rotating wheel. Samples were centrifuged at 4°C and >16,000 g for 1 hr and supernatants were used as nuclear protein extracts in the in vitro reconstitution DNA pull-down assays.

DNA pull-downs were performed as described (*Kappei et al., 2017*). Briefly, 25 µg of forward and reverse sequence oligonucleotides (*Supplementary file 4*) were diluted in annealing buffer (20 mM Tris-HCl, pH 7.5, 10 mM MgCl₂, 100 mM KCl), denatured at 95°C and annealed by cooling. Annealed double-strand oligonucleotides were incubated with 100 units T4 kinase (ThermoFisher) for 2 hr at 37°C followed by incubation with 20 units T4 ligase overnight. Concatenated DNA strands were purified using phenol-chloroform extraction. Following biotinylation with desthiobiotin-dATP (Jena Bioscience) and 60 units DNA polymerase (ThermoFisher) the biotinylated probes were purified using MicroSpin G-50 columns (GE Healthcare). DNA baits were immobilized on 500 µg paramagnetic streptavidin beads (Dynabeads MyOne C1, ThermoFisher) on a rotation wheel for 30 min at room temperature. Subsequently, baits were incubated with 400 µg of nuclear extracts from MCF7 cells in PBB buffer (150 mM NaCl, 50 mM Tris-HCl pH 7.5, 5 mM MgCl₂, 0.5% Igepal CA-630 [Sigma]) while rotating for 2 hr at 4°C. 10 µg sheared salmon sperm DNA (Ambion) was added as a DNA binding competitor. After three PBS washes (450/500/600 µL), bound proteins were eluted in 2× Laemmli buffer and boiled for 5 min at 95°C.

## Mass spectrometry data acquisition and analysis

DNA pull-down samples were separated on a 12% NuPAGE Bis-Tris gel (ThermoFisher) for 30 min at 170 V in 1× MOPS buffer (ThermoFisher). The gel was fixed using the Colloidal Blue Staining Kit (ThermoFisher) and each lane was divided into four equal fractions. For in-gel digestion, samples were destained in destaining buffer (25 mM ammonium bicarbonate; 50% ethanol), reduced in 10 mM DTT for 1 hr at 56°C followed by alkylation with 55 mM iodoacetamide (Sigma) for 45 min in the dark. Tryptic digest was performed in 50 mM ammonium bicarbonate buffer with 2 µg trypsin (Promega) at 37°C overnight. Peptides were desalted on StageTips and analyzed by nanoflow liquid chromatography on an EASY-nLC 1200 system coupled to a Q Exactive HF mass spectrometer (ThermoFisher). Peptides were separated on a C18 reversed-phase PicoFrit column (25 cm long, 75 µm inner diameter; New Objective) packed in-house with ReproSil-Pur C18-AQ 1.9 µm resin (Dr. Maisch). The column was mounted on an Easy Flex Nano Source and temperature controlled by a column oven (Sonation) at 40°C. A 105 min gradient from 2% to 40% acetonitrile in 0.5% formic acid at a flow of 225 nL/min was used. The spray voltage was set to 2.2 kV. The Q Exactive HF was operated with a TOP20 MS/MS spectra acquisition method per MS full scan. MS scans were conducted with 60,000 at a maximum injection time of 20 ms and MS/MS scans with 15,000 resolution at a maximum injection time of 50 ms. The raw files were processed with MaxQuant version 1.5.2.8 (*Cox and Mann, 2008*) with preset standard settings for SILAC labeled samples and the re-quantify option was activated. Carbamidomethylation was set as a fixed modification while methionine oxidation and protein N-acetylation were considered as variable modifications. Search results were filtered with a false discovery rate of 0.01. Known contaminants, proteins groups only identified by site, and

reverse hits of the MaxQuant results were removed and only proteins were kept that were quantified by SILAC ratios in both 'forward' and 'reverse' samples. Raw mass spectrometry data will be accessible through the ProteomeXchange Consortium via the PRIDE (*Vizcaíno et al., 2016*) partner repository with the dataset identifier PXD020473.

## Bioinformatics analyses

Primate genomes were accessed through Ensembl using the Compara application program interface via Bio::EnsEMBL::DBSQL::MethodLinkSpeciesSetAdaptor using the *method_link_type* 'EPO_LOW_-COVERAGE' and the *species_set_name* 'primates'. The human *A3B* gene, including the −900 to +50 region, was compared to other sequences and further analyzed using Geneious Prime software (version 2019.1.3). A3B promoter transcription factor binding sites were predicted using the JASPAR database (*Fornes et al., 2020*) with score threshold set to 80%.

TCGA primary breast tumors represented by both RNA-seq and whole-exome sequencing (*Cancer Genome Atlas Network, 2012*) were downloaded from the Firehose GDAC resource through the Broad Institute pipeline (http://gdac.broadinstitute.org/; n = 716). Genes correlated with *A3B* mRNA expression across the entire primary breast tumor data set were obtained using the USCS Xena Browser (bioRxiv 326470; doi: https://doi.org/10.1101/326470) and the top 20 genes in this list with the highest positive Spearman's correlations were used in this analysis. Quartiles for RSEM gene expression values relative to the housekeeping gene *TBP* were obtained for this gene list, and top (>75%) and bottom (<25%) quartiles of expression were calculated for every gene. Samples were then sorted into groups based on whether the expression of that gene fell into the top (n = 53 samples) or bottom (n = 111 samples) quartile for every gene in the list. RNA-seq data for *A3B* were also downloaded for every primary breast tumor, normalized to *TBP*, and used to establish expression correlations. The same methodology was used to calculate the top (>75%) and bottom (<25%) quartiles of expression for *A3B* mRNA to examine the mean *A3B* expression, APOBEC mutation signature, and APOBEC enrichment score.

APOBEC mutation signatures were determined as described (*Alexandrov et al., 2013*; *Jarvis et al., 2018*) using the deconstructSigs R package (*Rosenthal et al., 2016*). APOBEC mutation enrichment scores were calculated using the hg19 reference genome and published methods (*Chan et al., 2015*). Enrichment score significance was assessed using a Fisher exact test with Benjamini-Hochberg false discovery rate (FDR) correction. All dataset analyses and visualizations were conducted using R and the ggplot2 package (https://www.R-project.org/).

## Acknowledgements

We thank Yanjun Chen and Bojana Stefanovska for helpful comments, James DeCaprio for constructive feedback, Diako Ebrahimi for help analyzing promoter conservation, Walker Lahr and Brandon Moriarity for advice on HDR, Shuyu Meng for early ChIP contributions, and Jesenia Perez and Daniel Salamango for sharing preliminary data. The results presented here are in part based upon data generated by the TCGA Research Network: http://www.cancer.gov/tcga.

## Additional information

### Competing interests

Reuben S Harris: RSH is a co-founder, shareholder, and consultant of ApoGen Biotechnologies Inc. The other authors declare that no competing interests exist.

### Funding

| Funder | Grant reference number | Author |
| --- | --- | --- |
| National Cancer Institute | P01-CA234228 | Reuben S Harris |
| KWF Kankerbestrijding | KWF10270 | John WM Martens<br>Paul N Span<br>Reuben S Harris |
| National Medical Research | NMRC/OFYIRG/055/2017 | Dennis Kappei |

| Council | | | |
| --- | --- | --- | --- |
| National Research Foundation of Singapore | Centres of Excellence | Dennis Kappei |
| Ministry of Education - Singapore | Centers of Excellence | Dennis Kappei |

The funders had no role in study design, data collection and interpretation, or the decision to submit the work for publication.

### Author contributions

Pieter A Roelofs, Formal analysis, Investigation, Methodology, Writing - original draft, Writing - review and editing; Chai Yeen Goh, Data curation, Investigation, Methodology, Writing - review and editing; Boon Haow Chua, Matthew C Jarvis, Formal analysis, Investigation, Methodology, Writing - review and editing; Teneale A Stewart, Jennifer L McCann, Investigation, Methodology, Writing - review and editing; Rebecca M McDougle, Michael A Carpenter, Investigation, Writing - review and editing; John WM Martens, Paul N Span, Supervision, Funding acquisition, Methodology, Writing - review and editing; Dennis Kappei, Formal analysis, Supervision, Funding acquisition, Methodology, Writing - original draft, Writing - review and editing; Reuben S Harris, Conceptualization, Formal analysis, Supervision, Writing - original draft, Project administration, Writing - review and editing

### Author ORCIDs

Pieter A Roelofs ⬚ https://orcid.org/0000-0002-4921-7089
Teneale A Stewart ⬚ https://orcid.org/0000-0003-4837-9315
Jennifer L McCann ⬚ http://orcid.org/0000-0003-0458-1335
John WM Martens ⬚ https://orcid.org/0000-0002-3428-3366
Paul N Span ⬚ http://orcid.org/0000-0002-1930-6638
Dennis Kappei ⬚ http://orcid.org/0000-0002-3582-2253
Reuben S Harris ⬚ https://orcid.org/0000-0002-9034-9112

### Decision letter and Author response

Decision letter https://doi.org/10.7554/eLife.61287.sa1
Author response https://doi.org/10.7554/eLife.61287.sa2

# Additional files

### Supplementary files

- Supplementary file 1. Genes associating positively with *A3B* expression in breast cancer.
- Supplementary file 2. Log-transformed SILAC values from proteomics experiments.
- Supplementary file 3. Gene expression values relative to *TBP* mRNA in TCGA breast tumors.
- Supplementary file 4. Sequences of oligonucleotides used in this study.
- Transparent reporting form

### Data availability

Raw mass spectrometry data is accessible through the ProteomeXchange Consortium via the PRIDE (Vizcaino et al., 2016) partner repository with the dataset identifier PXD020473. Additional data generated or analysed during this study are included in the manuscript and supporting files.

The following dataset was generated:

| Author(s) | Year | Dataset title | Dataset URL | Database and Identifier |
| --- | --- | --- | --- | --- |
| Kappei D | 2020 | APOBEC3B promoter interactors | https://www.ebi.ac.uk/pride/archive/projects/PXD020473 | PRIDE, PXD020473 |

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
