## [Decision Letter]

[Editors' note: this paper was reviewed by Review Commons.]

**Acceptance summary:**

Although there have been previous studies into mechanisms that control APOBEC3B expression, the cis-elements and relative contributions of transacting factors that control of A3B expression are not well understood. In some types of solid tumors, (breast, lung, and bladder), it is not clear what "switches" on APOBEC expression during carcinogenesis. The author's findings represent an advance towards understanding the underlying mechanisms that control A3B transcription and how they can become dysregulated in tumors. As such, the findings are of interest to many researchers interested in the etiology of mutations in cancer and may be of interest to those wanting to understand how APOBECs are upregulated as part of the innate immune response.

---

## [Author Response]

Reviewer #1 (Evidence, reproducibility and clarity):In the manuscript, "The RB/E2F pathway controls expression of the cancer genomic DNA deaminase APOBEC3B3," Roelofs et al. present a step-by-step examination of the mechanisms by which the RB/E2F pathway controls APOBEC3B expression. They begin by identifying probable E2F binding sites and determining the contribution of these sites to controlling A3B expression using an A3B-pomoter-luciferase reporter. They identify and further characterize the one site that is required for repression of the A3B promoter, identify a close by CHR element that is also required for repression, and subsequently use CRISPR to generate mutations in this site in the genome of MCF10A cells, which alleviates repression of the A3B promoter and increases A3B expression. They use SILAC to identify proteins that bind this specific E2F binding site and identify the DREAM complex and PRC1.6 complex as likely regulators. The authors then use either siRNA or overexpression of E2F4 or E2F6 to show the DREAM complex and the PRC1.6 complex repress A3B expression in MCF10A cell or BT-474. The authors also present evidence that A3B may be regulated along with other genes whose expression is regulated by E2F. The authors employed a diverse set of (often technically challenging) methods and (in most cases) produced convincing evidence to support their conclusions. Determining the cis-elements and transacting factors that regulate A3B expression is clearly important for both cancer biology and an understanding how APOBECs get upregulated in response to viruses. However, several additional small experiments are needed to support some of the author's conclusions. Also, there is a major concern about the significance of the findings, which the authors need to address.

Thank you for the accurate and positive synopsis of our work and for recognizing the use of a “diverse set of (often technically challenging) methods” to produce “convincing evidence”. As detailed below, we have addressed your major concern about significance and have also provided additional supporting results and clarifications.

Major concern(s):The authors of this study present a very detailed examination of the cis elements required for regulation of E2F regulation of A3B expression and show that the PRC1.6 complex can also repress A3B expression, which are novel results. However, the regulation of A3B by RB/EF2, and specifically repression by E2F4, acting as a member of the DREAM complex, is not a new finding, which lessens the significance of this study. Results presented in Periyasamy et al., 2017, show that p53 regulation of APOBEC3B expression is mediated by the E2F4/RB-containing DREAM repressive complex. This previous publication showed that repression of A3B expression (in colon cancer cell line HCT116) depends on E2F4 expression. They also demonstrated that recruitment of DREAM complex members E2F4, LIN9, and p130 to the A3B promoter (in breast cancer cell line ZR-75-1) increases upon p53 activation and that indicators of A3B activity are reduced by p53 activation (likely due to repression by the DREAM complex). They also show that the presence of DREAM complex members at the A3B promoter is reduced by expression of HPV E7 and E6. The fact that the DREAM complex reduces A3B expression in different contexts (in this earlier study) indicates this is a common mechanism to control A3B expression that likely functions in normal and cancer cells. The authors should acknowledge these previous findings in their Introduction. Although defining the cis elements that regulate gene expression, in this case specifically A3B, is an important and perhaps understudied aspect of APOBEC regulation, the broader significance of the cis element characterized in this study (by itself) is questionable given previous identification of the DREAM complex as a negative regulator of A3B expression. Before this paper is considered for publication, the authors should also elaborate on how the results of this study (1) are novel because they differ from these previous findings and/or (2) represent a significant advance in knowledge beyond the previous study. Alternatively, the authors should, (3) perform some limited additional experiments to advance the novel conclusions of this study.

Thank you for appreciating the big picture that “determining the cis-elements and transacting factors that regulate A3B expression is clearly important for both cancer biology and an understanding how APOBECs get upregulated in response to viruses”, and also for appreciating the fact that *the vast majority of our work is novel* including the delineation of the critical E2F binding site in the *A3B* promoter and the identification of a repressive role for the E2F6-containing PRC1.6 complex. Moreover, to reach these conclusions novel episomal (transient) and chromosomal (stable) systems had to be developed for studying *A3B* transcriptional regulation, which are reported for the first time here and we hope will drive future work in this area.

We did recognize the prior publication by Periyasamy and co-workers, which was cited in our original Discussion. However, we didn’t want to draw too much attention to this work because we have been unable to reproduce their key result that p53 represses A3B (which calls to question many of their results). First, in collaboration with Prof. Chris Lord’s and Prof. Alan Ashworth’s groups, we have demonstrated that CRISPR-mediated knock-out of p53 has no effect on endogenous *A3B* transcription in 293 cells (Nikkila et al., 2017). This negative result is not likely to be cell-type specific because, second, we found that CRISPR-mediated knock-out of p53 has no effect on *A3B* transcription in MCF10A (in the presence or absence of endogenous *A3B* upregulation by PMA) and, third, that treatment of MCF10 and MCF7 cells with the p53 inhibitor nutlin had no effect on *A3B* transcription at the same time as inducing expression of control genes (Starrett et al., 2019). Thus, in contrast to Periyasamy et al., we are confident that p53 has no role in *A3B* transcription (despite the fact that we were the first to hypothesize such a role in Burns et al., 2013 and Vieira et al., 2014). Rather, the vast majority of published data indicate that p53 more likely functions *downstream* of *A3B* expression by mediating DNA damage responses to A3B-catalyzed DNA deamination. Additionally, even though we originally also observed a modest increase of A3B protein expression upon E2F4 depletion in MCF10A, this observation did not hold up after additional rounds of biologically independent knockdown experiments.

Nevertheless, despite our general inability to reproduce several of the key findings by Periyasamy et al., we have revised our manuscript to include two additional citations to their 2017 paper in Nucleic Acids Research. First, the following lines are included in the Results section: “Although prior work has implicated the E2F4/DREAM complex (Periyasamy et al., 2017), the potential involvement of E2F6/PRC1.6 is novel.” Second, we have revised the Discussion to summarize the results from both our group and theirs with the E2F4/DREAM complex: “Although prior studies implicated the E2F4/DREAM complex and generally the RB/E2F axis in repressing A3B transcription (Periyasamy et al., 2017, Starrett et al., 2019), the work here is the first to define the responsible cis-elements (juxtaposed CHR and E2F sites), show that all PyV tTAg-mediated activation occurs through this single bipartite sequence, and demonstrate coordinated repression by not only the E2F4/DREAM complex but, surprisingly, also by the E2F6/PRC1.6 complex. […] The role of p53 in *A3B* transcriptional regulation is less clear with some studies indicating that p53 inactivation leads to A3B upregulation (Menendez et al., 2017, Periyasamy et al., 2017) and others demonstrating that *p53* knockout has no effect on *A3B* transcription (Nikkila et al., 2017, Starrett et al., 2019).”

Our Discussion has been additionally revised to highlight novel findings with the

E2F6/PRC1.6 complex including proteomics experiments showing binding to *A3B* promoter DNA in vitro (Figure 4), ChIP experiments demonstrating binding to *A3B* promoter DNA in living cells (Figure 5A-C), knockdown studies showing elevated *A3B* expression (Figure 5D), and overexpression studies leading to *A3B* repression in multiple breast cancer cell lines (Figure 5E). Moreover, additional new results in Figure 3F-G demonstrate that all PyV TAg-mediated derepression of *A3B* occurs through the single E2F site at +21-28 in the *A3B* promoter, and that this axis and the PKC/ncNF-κB pathway independently regulate endogenous *A3B* expression. Altogether our results support a novel model in which *A3B* expression is repressed throughout the cell cycle by the concerted activities of both the E2F4/DREAM and E2F6/PRC1.6 complexes (Figure 7). It is also worth specifically highlighting the double E2F4/6 knockdown experiment in Figure 5D, which demonstrates that both the DREAM and PRC1.6 complexes combine to repress *A3B* expression and also that E2F6 may play the dominant role.

Additional comments and thoughts:1) The authors present an interesting model (Figure 7) in which the DREAM complex (represses A3B expression during G0 and early G1) and trades off with the PRC1.6 complex (represses A3B expression during late G1 and S), but little experimental evidence supporting this switch. Considering that the repressive activity of the DREAM complex is more often reduced during cancer (by oncogenic signaling), it seems possible that in some cell contexts, A3B expression could fluctuate during the cell cycle and be lower during S-phase. Consequently, A3B expression may be less during DNA replication when the potential to introduce mutations is the highest. This also might mean A3Bs mutagenic potential may be over-estimated. It would be interesting to determine if A3B protein levels fluctuate with the cell cycle in cells with reduced DREAM complex function. The authors could also employ either ChIP or SILAC experiments on synchronized cells to support their "repressor switch" model. This would constitute a more significant finding.

We thank you for taking the time to provide additional comments. As described above, additional experiments including ChIP forced us to revise the model in Figure 7 to repression of *A3B* by the combined activity of E2F4/DREAM and E2F6/PRC1.6 complexes.

2) The authors focused solely on the repressive E2F signaling. The authors are well situated to determine if E2F activating transcription factors have any impact on A3B expression. The authors could also do some experiments to investigate the relationship between activating and repressive E2F factors in control of A3B expression. Several specific ideas: The authors could consider if mutations to E2F binding site D, which decrease expression from the A3B promoter, is a binding site for activating E2Fs or if this site overlaps other transcription factor binding sites. The authors should also determine if E2F binding site E, is bound by activating E2Fs when repressive factors E2F4 and E2F6 are depleted. This is one plausible explanation that could explain why A3B expression is less in MCF10A cells with the E2F (E) element deleted compared to BT-474. Are activating E2Fs associated with A3B promoter and are they increased by decreasing E2F4 and E2F6. Addressing these ideas would make for a much more impactful manuscript.

We appreciate all of these points and have added the following experiments to the manuscript. First, we show that tTAg overexpression causes upregulation of endogenous A3B in MCF10A cells with a wildtype *A3B* promoter (as expected) but not additional upregulation of *A3B* expression in the site E mutant constructed by HDR (Figure 3F). This result is important by indicating that tTAg induces *A3B* by removing repressive E2F complexes and, simultaneously, that tTAg is unlikely to be inducing activating E2Fs to work at other promoter proximal E2F sites such as site D.

Second, although not specifically requested, we provide the following comparative experiment to demonstrate the complexity and elegance of *A3B* transcriptional regulation. We demonstrate that PMA induction of A3B through the PKC/non-canonical NF-κB axis (as we reported originally, Leonard et al., 2015) causes even higher levels of *A3B* expression in the MCF10A E2F site E mutant (Figure 3G). This result contrasts with the epistasis observed with tTAg overexpression and strongly indicates that the PKC/ncNF-κB and RB/E2F4/E2F6 pathways independently combine to regulate *A3B* gene expression.

3) In the future, the authors should consider investigating if transcription factor binding sites exist in intron 1 of the A3B gene. This is a region often contains cis-regulatory elements/transcription factor binding sites. This is just a helpful suggestion and does not need to be addressed.

Again, this is a great idea and we plan to address it more in future work (note that one of three NF-κB binding sites implicated by our prior ChIP and bioinformatic studies is located in intron 1 – Leonard et al., 2015).

Reviewer #1 (Significance):As a researcher interested in the etiology of APOBEC-signature mutations in cancer, I found this study interesting. There is little doubt that A3B is involved in multiple aspects of cancer biology. Although other APOBECs also contribute to mutagenesis in cancer, A3B certainly generates APOBEC signature mutations in some tumors. A3B also plays major roles in innate immunity to restrict replication of a subset of viruses. Although there have been previous studies into mechanisms that control A3B expression, the cis-elements and relative contributions of transacting factors that control of A3B expression are not well understood. In some types of solid tumors, (breast, lung, and bladder), it is not clear what "switches" on APOBEC expression during carcinogenesis. The author's findings represent an advance towards understanding the underlying mechanisms that control A3B transcription and how they can become dysregulated in tumors. As such, the findings would be of interest to many researchers interested in the etiology of mutations in cancer and may be of interest to those wanting to understand how APOBECs are upregulated as part of the innate immune response. However, unless I am missing something, I believe the significance of the manuscript needs to be strengthened by taking one of the novel findings one small step further.

This synopsis of the field and importance of A3B is very good. As described above in response to your major concern and in response to several of your additional points, we report several novel findings including the exact delineation of the E2F binding site within the *A3B* promoter, both a non-biased and a data-guided approach in identifying key regulatory complexes at this E2F site, and the concerted effort of both DREAM and PRC1.6 complexes in *A3B* repression. The significance of the work is further strengthened by additional experiments showing that the identified E2F biding site is the only site required for repression of *A3B* by these two complexes, and that *A3B* expression can be further enhanced through the independently functioning PKC/non-canonical NF-κB axis. Additionally, we show that using a E2F-deregulated gene set is a better indicator of APOBEC mutagenesis than *A3B* expression alone. In short, this is the first work to demonstrate that A3B is a dual E2F4/DREAM- and E2F6/PRC1.6-repressed gene and establish an unambiguous linkage to similarly regulated cell cycle genes. Together with a similarly strong mechanistic linkage to PKC/(nc)NF-κB

(inflammation and immunity), it is becoming clear that multiple signal transduction pathways can converge, in many instances simultaneously, to promote A3B expression and mutagenesis in cancer (the “perfect storm” described in the final line of our revised Discussion).

Reviewers cross-commenting:I agree with the second reviewer that the analysis in Figure 6 could be improved and that in general the work in the manuscript is well done. My major concern with the work, however, is the extent of novelty as the p53 and the DREAM complex have previously been shown to regulate A3B expression. I feel that expanding how the PRC1.6 complex regulates A3B expression, potentially in a cell cycle regulated manner is important to increase the significance of the findings.

Please see responses above. We hope you agree that we have done much more to understand the involvement of the E2F6/PRC1.6 complex in regulating *A3B* expression.

Reviewer #2 (Evidence, reproducibility and clarity):This manuscript describes a series of investigations regarding the control mechanism of APOBEC3B (A3B) expression. A series of sequence investigations and molecular biology experiments established that the presence of an E2F and a CHF site at the A3B promoter contributes to repress A3B expression. Proteomics data confirm the involvement of E2F4 and E2F6-involving complexes in such repression, which was also supported by some bioinformatics analysis.In my opinion this manuscript highlights convincingly the importance of the E2F proteins in regulating A3B expression from both the genetic and the proteomics perspectives. I feel that the bioinformatics analyses need to be strengthened to substantiate some of the claims shown otherwise elegantly in the experiments on cell lines.

Thank you for your thoughtful and positive comments. As requested, we have strengthened the bioinformatics analyses in the revised manuscript as described below.

Major comments:1) A question left open relates to the relative importance of E2F4 vs E2F6 is repressing endogenous A3B expression – the authors show experimentally that both are relevant. However, there are conflicting evidence presented – that E2F4 overexpression de-represses A3B but endogenously E2F4 is cytoplasmic – could the authors comment on this further?

We are not completely sure what you mean here as our model supports repression (and not de-repression) of *A3B* by E2F4 and E2F6 complexes. However, new knockdown and ChIP experiments (revised Figure 5) indicate that E2F6 is more important in repressing *A3B* transcription in MCF10A cells. Most importantly, we have repeated knockdown studies in MCF10A and show that the E2F6 single knockdown has a greater effect than the single E2F4 knockdown and that the E2F4/E2F6 double-knockdown causes a level of *A3B* de-repression (expression) that is greater than either single-knockdown alone (Figure 5D). The more-than additive increase with the combined knockdown indicates that these two repressive complexes have an at least partly overlapping function (model in Figure 7). We also present new ChIP data showing that the E2F binding site can be occupied by either E2F4 or E2F6-based repressive complexes (Figure 5A-C), again emphasizing the concerted effort of both complexes in repressing *A3B.* These experiments have made the IF panel somewhat superfluous. In an effort to keep our current manuscript to the point (and to allow space for more functionally relevant data) we opted to omit the localization data from the revised manuscript.

Are E2F4/E2F6 over-expressed in tumours? If one looks into A3B co-expressing genes in tumours, which complex is more positively correlated with A3B expression in tumours – E2F4 or E2F6? In the Discussion the authors mention the hand-off model – this suggested analysis might help in either supporting (if you find there is no difference in terms of which one is more correlated with A3B) or refuting (otherwise, you find a principal E2F protein involved in its regulation) this hypothesis.

This is an interesting thought but based on our mechanistic studies here we would actual predict the opposite – that lower levels of repressive E2F4 and/or E2F6 might correlate with higher levels of A3B. However, prior studies have shown that *E2F4* and *E2F6* mRNA levels do not change significantly through the cell cycle (Cuitino et al., 2019). Moreover, in separate analyses of TCGA data sets, we have found no major difference in mRNA levels of either *E2F4* or *E2F6* in normal mammary tissue versus primary breast tumors (mean mRNA levels relative to *TBP*: *E2F4* is 3.2 in normal and 3.4 in tumors, and *E2F6* is 1.3 in normal and 1.4 in tumors).

2) The bioinformatics analyses were very concisely described and does show a relation between A3B, E2F and APOBEC signature enrichment, which is an encouraging piece of result. However, I feel this analysis needs to be unpacked further and structured to back up some of the other claims presented earlier in the paper. E.g. If one does a Gene Set Enrichment Analysis (GSEA) on the A3B co-expressing genes, are E2F4 or E2F6-repressed genes enriched?

We have revised this part of the Results section to better explain our rationale, approach, and results. We have also added an additional A3B-focused, quartile-based analysis for comparison (Figure 6E-H). GSEA and IPA were not particularly informative because, as explained above, these two genes are not regulated differentially throughout the cell cycle. For instance, the GSEA associations were general and not informative mechanistically (top 2 were mitotic nuclear division and chromosome segregation) and, therefore, we would prefer not to include these analyses in the manuscript.

3) I also feel the binning of samples by quartile etc. as the authors did, was rather crude. Are there analogous comparisons treating gene expression is a continuous variable instead of the arbitrary bins presented here? E.g. if one scales gene expression (say z-transform or equivalent) and performs GSEA on this (against the E2F gene set that the authors compiled/acquired), isn't this a more all-rounded statistic to quantify A3B (and E2F) activity – with which the association with APOBEC signature enrichment etc. could be assessed?

We have done the regression analysis as requested and the overall result is similar. We can include this additional analysis in the manuscript if the reviewer wishes. However, we would rather include an additional analysis showing that overexpression the 20-gene set of functionally E2F-repressed genes associates more strongly with APOBEC mutation signature high tumors than overexpression of *A3B* mRNA alone (compare Figure 6A-D vs Figure 6E-H). We suspect this 20-gene set enables the identification of tumors that are somehow more sensitive to (or tolerant of) the APOBEC mutagenesis program but many more studies will be necessary to fully explore this exciting possibility.

Reviewer #2 (Significance):This manuscript provides good and comprehensive evidence on the regulation of A3B expression. This could have important implications towards controlling the expression of this important gene in carcinogenesis. The experimental data are well-rounded, comprising both molecular/cell biology and quantitative proteomics experiments. This extends existing literature on APOBEC3 gene expression by presenting additional pathways of regulating its expression. This is likely to interest molecular and cell biologists interested in gene regulation, as well as cancer biologists interested in APOBEC3 mutagenesis.

Thank you very much for this clear synopsis and for recognizing the broader impact of our studies.